# Areas of endemism of land planarians (Platyhelminthes: Tricladida) in the Southern Atlantic Forest

**Domingo Lago-Barcia**[1,2]*, **Marcio Bernardino DaSilva**[3], **Luis Americo Conti**[4],
**Fernando Carbayo**[1,2]

**1** Laboratório de Ecologia e Evolução, Escola de Artes, Ciências e Humanidades (EACH), Universidade de São Paulo (USP), São Paulo, SP, Brazil, **2** Departamento de Zoologia, Instituto de Biociências, Universidade de São Paulo (USP), Rua do Matão, São Paulo, SP, Brazil, **3** Departamento de Sistemática e Ecologia, CCEN, Universidade Federal da Paraíba, Cidade Universitária, Conj. Pres. Castelo Branco III, João Pessoa, PB, Brazil, **4** Escola de Artes, Ciências e Humanidades (EACH), Universidade de São Paulo (USP), São Paulo, SP, Brazil

* domingo.lagobarcia@gmail.com

## Abstract

Areas of endemism (AoE) are the main study units in analytical biogeographic methods, and are often defined as an area with two or more endemic species living in them, presenting substantial congruence among their range limits. We explored the distribution of land planarians (Geoplanidae, Platyhelminthes) across the southern region of the Brazilian Atlantic forest (from the state of Rio de Janeiro, to the state of Rio Grande do Sul) utilizing DaSilva's *et al.* (2015) protocol. We used two methods, Endemicity Analysis (EA), and Geographical Interpolation of Endemism (GIE). We identified nine AoE of terrestrial flatworms in the Southern Atlantic forest. Performance of the methodologies is discussed. These AoE of land planarians can be explained through vicariance events combined with their physiological and ecological own limitations. Interestingly, these AoE are congruent with fine-scale approaches such as that with harvestmen. Most land planarians have revealed to present a very small distributional range evidencing their potential as a good model for fine-scale studies of AoE.

## Introduction

Historical biogeography has evolved substantially since its beginning three centuries ago [1]. One of its main objectives is to detect areas of endemism (AoE) [2]. Because AoE are the main study units in analytical biogeographic methods [3, 4], they can serve to design biogeographic regionalization schemes [5], to infer historical relationships between them [6], to study organism-climate dynamics [7], and as a criterion to identify areas for conservation [8, 9, 10, 11]. Different methodologies have been proposed to achieve said objectives [12–21] with different animal taxons [3, 22–26].

AoE are hypothesized areas that can vary depending on the data use [13, 27–30]. AoE are often defined as an area with two or more endemic species living in them, presenting

**Funding:** The present work was financed with support from Fundação de Amparo à Pesquisa do Estado de São Paulo (FAPESP) (Proc. 2016/18295-5) (http://www.fapesp.br/) and a Graduate Fellowship from Conselho Nacional de Desenvolvimento Científico e Tecnológico (CNPq) (http://www.cnpq.br/). The funders had no role in study design, data collection and analysis, decision to publish, or preparation of the manuscript.

**Competing interests:** The authors have declared that no competing interests exist.

substantial congruence among their range limits [27]. Those basic criteria are based on the concept of a common biogeographical origin and/or isolation of those species ranges [3, 31].

The Atlantic Forest is one of the largest forests in the Americas, extending over approximately 150 million hectares [32]. Longitudinal, latitudinal and altitudinal ranges have modeled different climatic and environmental conditions producing distinct environments in it [33, 34]. These different conditions have created a complex biome that is characterized by high levels of endemism (averaging nearly 50% overall, and as high as 95% in some groups; [35]).

Since the proposition of the first biogeographic regionalization of the world, by Wallace [36], many studies identifying AoE have been carried out, including those in the Atlantic Forest in the 20th century (Fig 1). Through the years, the AoE proposed for the Atlantic Forest vary in size and number depending on the taxonomic group being studied. Müller [22] divided the Atlantic Forest into 3 AoE using amphibians, reptiles, birds and mammals. He named them Pernambuco, Bahia and Paulista. The author considered AoE as those areas where endemic species distributions overlapped. Kinzey [23] identified the same AoE as Müller, though smaller in size, using primate distributions. Cracraft [3] suggested two AoE using bird distributions; one of these areas extends from Pernambuco to Santa Catarina, and the other covers the so called "Paranaense Forest", as named by Morrone [31]. Amorim & Pires [37] detected two large AoE, using diptera and primate distributions. These two large areas are located in the borderline between Espírito Santo and Rio de Janeiro states and were subdivided into smaller areas, namely the northern area, in turn with four subareas, and the southern area, with two. Costa *et al.,* [38] also discovered two large AoE using mammals distribution. Differently from Amorim & Pires [37] and Müller [22], they positioned the limit between these two areas 300 km further south, though, concordant with Müller [22], the northern area was subdivided into the same four subareas. Silva & Casteleti [24] detected five AoE of butterflies, birds and primates. Those areas were named Brejos Nordestinos, Pernambuco, Diamantina, Bahia and Serra do Mar. Brejos Nordestinos are higher altitude forest spots, surrounded by Caatinga in the states of Ceara, Pernambuco, Paraiba, and Piaui states; Pernambuco is situated in the coastal region of occidental Northeast Brazil; Diamantina is located in the interior of the state of Bahia; Bahia extends from the state of Sergipe to the state of Espirito Santo; Serra do Mar comprises the mountainous areas from Rio de Janeiro to Rio Grande do Sul states. The five AoE of Silva & Castelei [24] were delimited adjacent to three mixed transitional areas named São Francisco, Florestas do Interior and Florestas de Araucária. Silva *et al.* [39] proposed four AoE using bird ranges: Pernambuco (as defined in previous studies); Bahia Central (corresponding to Diamantina in Silva & Casteleti [24]); Bahia (homonym of Bahia AoE in Silva & Casteleti [24]); and Serra do Mar, the latter located on both sides of the Rio Doce, and extending from Espírito Santo to the northern Santa Catarina. Prado *et al.* [26] detected a single AoE of rodents in the Atlantic Forest region, the so-called Eastern South America, and subdivided it into four subregions. Each subregion would coincide with Pernambuco, Bahia, Diamantina and Serra do Mar AoE of Silva & Casteleti [24], respectively. Pinto-da-Rocha *et al.* [40], DaSilva & Pinto-da-Rocha [41] and DaSilva *et al.* [25] detected 12 AoE using harvestmen distributions. Only two out of the 12 were coincident with previously proposed AoE, namely Pernambuco and Bahia. The remaining ten AoE are small areas placed within AoE mentioned in the literature. Hoffmeister *et al.* [19] detected 29 groups of AoE. Some of these groups were congruent with previously delimited areas discovered by different authors. Based on orchid bee distributions, Garrafoni *et al.* [42] found at least five main AoE, which are congruent with those from Silva & Casteleti [24], varying only in size.

Refinement of the methods in the discovery of AoE has accordingly produced more refined, congruent areas inferred from different taxonomic groups. General differences are related to size and shape of the areas.

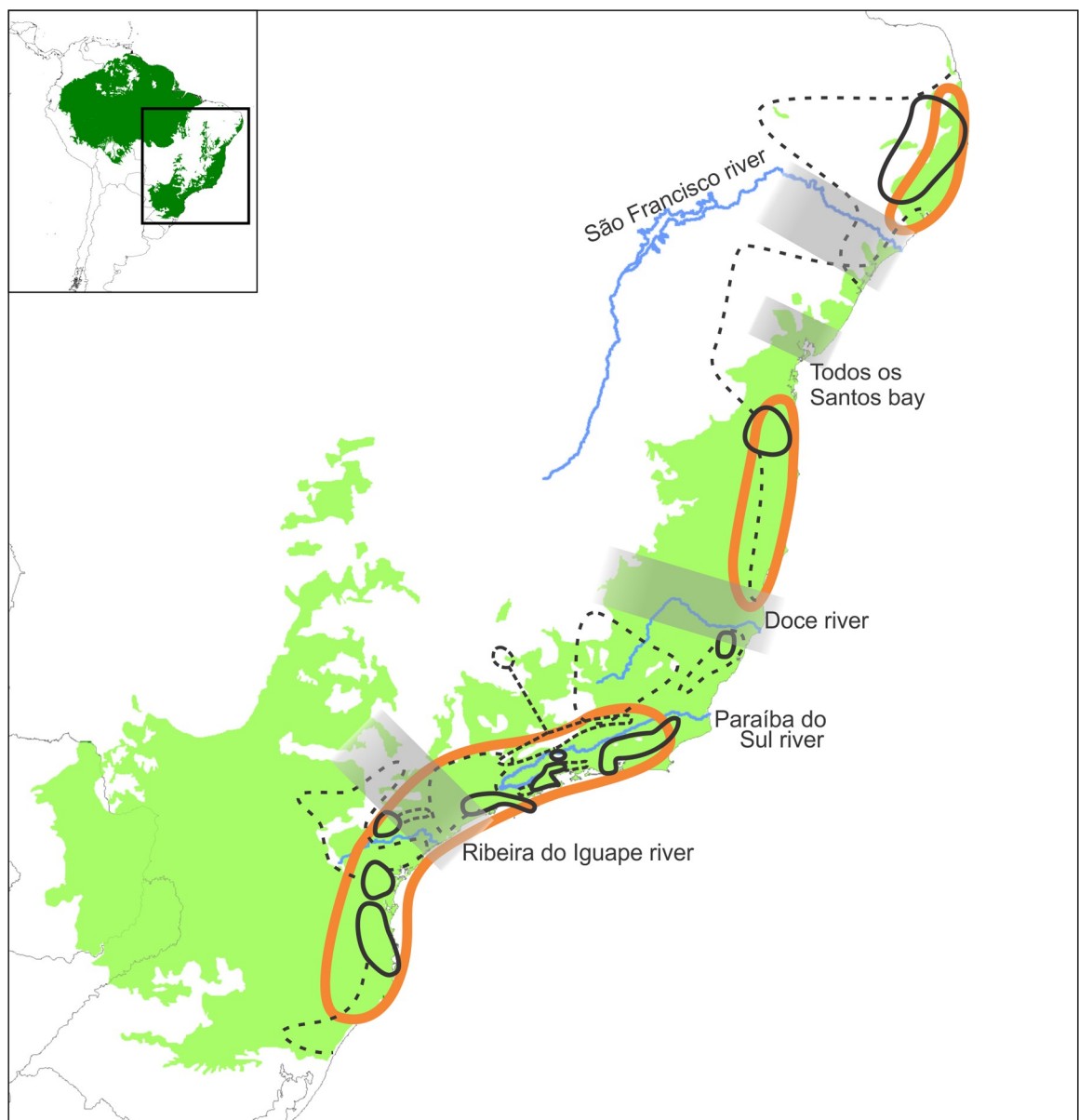

**Fig 1. General spatial pattern of historical breaks of Atlantic forest for several taxa.** In wide orange lines, congruence of AoE of most works in literature (north to south, Pernambuco, Bahia and Serra do Mar AoEs, see references in the text); in black lines, areas of endemism for harvestmen ([25], full line is Congruence Cores, pointed black line is Maximum Region of Endemism). Gray bands represent main divisions as inferred from phylogeography, panbiogeography, cladistic biogeography and systematics of many works in literature (see S1 Material).

Numbers of AoE have varied according to progressive refinement of the methods and the taxonomic group used as model, either animals or plants. It is expected that lineages coexisting in a geographical space will also share overlapped biogeographic history. Deviations from this situation may be assigned to contingencies and to the unique ecological and physiological constraints undergone by each lineage. By contributing with additional taxa, new distributional data and new historical reconstructions of unexplored taxonomic groups, progress can be achieved in AoE knowledge [31] and new AoE hypotheses can be generated. Terrestrial

flatworms or land planarians (Platyhelminthes, Geoplanidae) are one of those unexplored taxonomic groups. No biogeographic studies have been previously done using these animals. Terrestrial flatworms are very sensitive to changes in the conditions of their environment [43], are highly endemic due to their low vagility and dependence on humidity [44, 45] and are endemic to relatively small areas [46]. Furthermore, recent taxonomic revisions revealed that land planarians have smaller distributions than what was previously thought, due to the discovery of cryptic species under the same name [47–49]. For these reasons, land planarians are considered an optimal taxonomic group to carry out biogeographic studies. This paper aims to contribute to the debate of discovery and test of congruence of areas of endemism in the Atlantic forest. For this purpose, we explored the distribution of land planarians across the Atlantic Forest using two methods, Endemicity Analysis (EA), and Geographical Interpolation of Endemism (GIE).

## Materials and methods

### List of abbreviations

AoE: Area of endemism

CC: Congruence Core

EA: Endemicity Analysis

GIE: Geographic Interpolation of Endemism

MIS: Area of endemism of Misiones

MRE: Maximum Region of Endemism

NDM/VNDM: Software that carries out an Endemicity Analysis

NSC: Area of endemism of North Santa Catarina

ORG: Area of endemism of the Serra dos Órgãos

POA: Area of endemism of Porto Alegre

PR: Area of endemism of Paraná

SFP: Area of endemism of São Francisco de Paula

SMSP: Area of endemism of Serra do Mar de São Paulo

SSC: Area of endemism of South Santa Catarina

SSP: Area of endemism of Southern São Paulo

WS: Widespread

The study area comprises the areas covered with Atlantic forest between, and including, the state of Espírito Santo (the northernmost) and Rio Grande do Sul (the southernmost). We compiled distributional data belonging to 270 species of geoplanids (Tables 1 and S1). For these species, 403 records were obtained from the literature and 166 from our own data, summing up a total of 570 records in our dataset (S1 Table and S2 Material). Species with doubtful records or identification, from literature or our own data, were not considered (S2 Table). One record is here understood as an observation of a species (either from literature or from our own data) in a certain locality, independently from the number of times this species was found.

**Table 1. The nine Areas of Endemism (AoE) discovered through Endemicity Analysis (EA) and/or Geographic Interpolation Endemism (GIE).**

| AoE | EA | GIE | Endemic spp. scoring for a CC | Endemic spp. in the CC represented by a Single record | Endemic spp. in a CC | Spp. contributing to the MRE | Endemic spp. (exclusive to the AoE) | Widespread spp. |
|---|---|---|---|---|---|---|---|---|
| ORG | 0.25, 0.5 | Yes | 2 | 26 | 30 | 2 | 32 | 18 |
| SMSP | 0.25, 0.3, 0.5 | Yes | 3 | 15 | 18 | 6 | 24 | 44 |
| SSP | - | Yes | - | 15 | 15 | 3 | 18 | 13 |
| PR | - | Yes | - | 10 | 10 | 1 | 11 | 9 |
| NSC | 0.1, 0.25 | Merged with SSC | 2 | 10 | 12 | 0 | 12 | 18 |
| SSC | 0.1, 0.25, 0.3 | Merged with NSC | 2 | 18 | 23 | 0 | 23 | 16 |
| SFP | 0.1, 0.25, 0.3 | Merged with POA | 5 | 20 | 25 | 0 | 25 | 13 |
| POA | 0.5 | Merged with SFP | 3 | 1 | 4 | 0 | 4 | 12 |
| MIS | 1.0 | Yes | 2 | 4 | 6 | 0 | 6 | 0 |

For EA, the cell sizes through which EA discovered AoE are indicated. The Table also informs the number of flatworm species according to different classification criteria. CC: Congruence Core; MRE: Maximum Region of Endemism. See text for details.

For our own data, we intensely sampled land planarians in six areas (Fig 2) (Parque Estadual de Intervales, Parque Nacional de Saint-Hilaire/Lange, Parque Nacional da Serra do Itajaí, Parque Estadual da Serra do Tabuleiro, Parque Nacional de São Joaquim and Floresta Nacional de São Francisco de Paula; ~200 person-hour of sampling). Other localities were sampled sporadically between 2009 and 2018. Each of these intensely sampled areas is at a distance of ~120 km in a straight line from the next one. As a result, we collected 1621 specimens. These specimens were diagnosed by either only their external aspect (for known and characteristic species regarding size, shape and color of the body) or through external aspect and morphological details of the cephalic region, the pharynx and the copulatory apparatus examined on histological glass slides produced following Carbayo & Almeida [50] (S1 Table). 156 out of the 1621 specimens could not be identified either because of their poor state of conservation or due to their immature developmental status and were not further considered in this study. The remaining 1465 specimens belong to 94 known and 98 undescribed species, totalizing 192 species (S1 Table). We used GPS device Garmin E-Trex for recording the geographic coordinates with a maximum error of ~50 m of most of the specimens we collected. We used Googlemaps for recovering geographic coordinates of locations mentioned in the literature and a few of our own specimens which could not be georeferenced.

To discover AoE, the dataset (S2 Material) was analyzed using two methods: (i) the Endemicity Analysis (EA), which uses a heuristic algorithm of the software NDM/VNDM (version 3.1, developed by Goloboff [51], Szumik et al. [14] and complemented by Szumik & Goloboff [15]), and (ii) the Geographical Interpolation of Endemism (GIE) based on a Kernel interpolation [18]. Resulted patterns were evaluated by combined criteria by DaSilva et al. [25], whose protocol was applied for a final delimitation.

The first approach (i), EA, is based on an optimality criterion which uses species distributions to identify AoE. It uses an heuristic algorithm to calculate an endemicity score for each set of cells containing endemic species on a grid on a map. The criteria used in this method were: (a) an area consisting of two or more grid cells presenting two or more endemic species (any species whose range is restricted to these or adjacent grid cells; see below), and (b) an endemicity score above two in each grid cell. We adjusted the points of grid origin to the

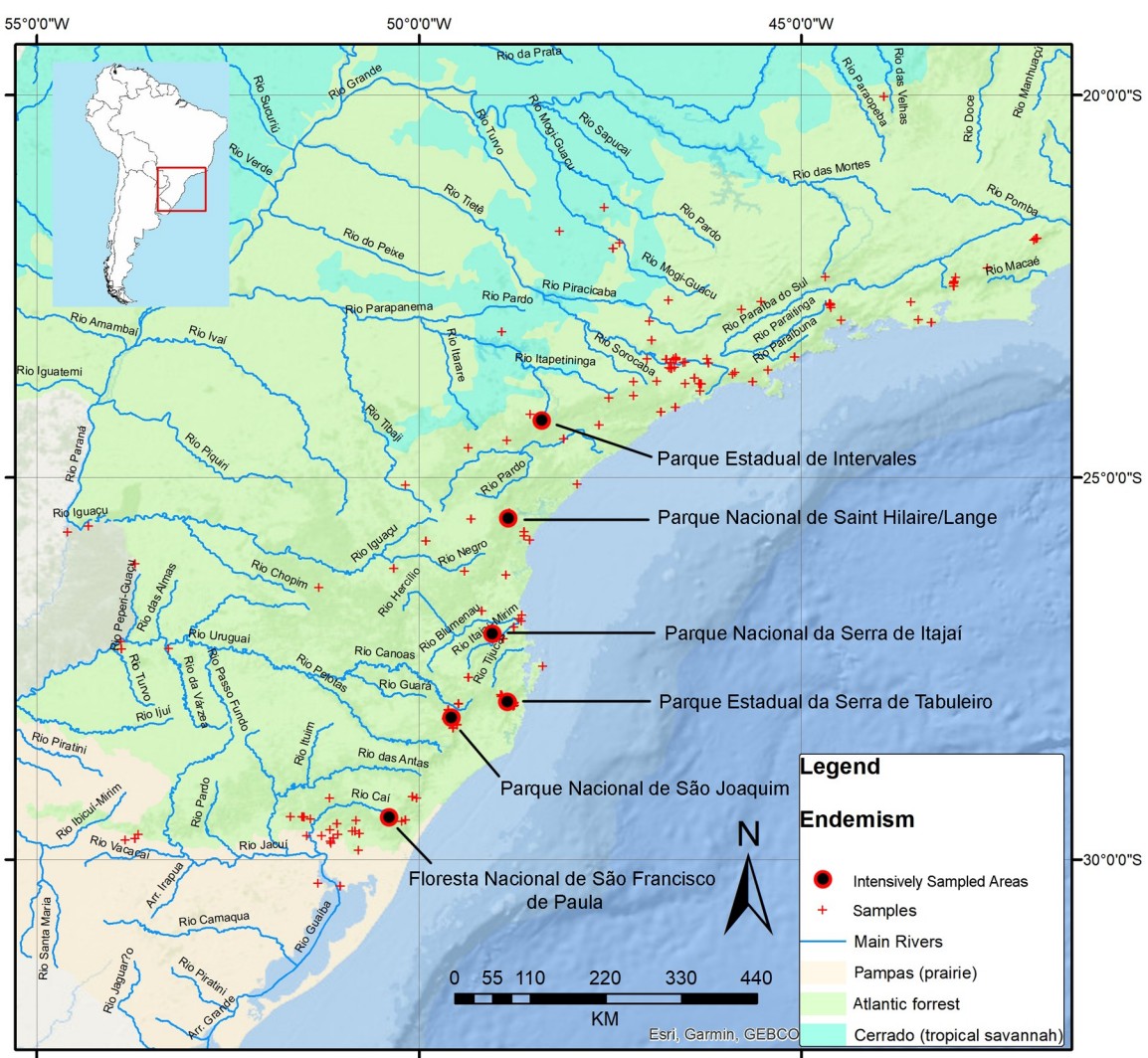

**Fig 2. Localities intensively sampled (~200 person-hour of sampling) in the Atlantic forest.** Basemap source: Esri. "Topographic" [basemap]. Scale Not Given. "World Topographic Map". https://www.arcgis.com/home/item.html?id=30e5fe3149c34df1ba922e6f5bbf808f.

default position provided by NDM and to three other origins randomly selected to test for eventual hidden AoE caused by its position, and subsequently run 100 replicas with each of five different grid cell sizes (0.1˚, 0.25˚, 0.3˚, 0.5˚, and 1˚) to examine the effect of cell size on inferred patterns of endemism. We largely follow the protocol proposed by DaSilva *et al.*, [25] for several reasons discussed further below (see section *Discussion: 1. Methodological observations*). We computed consensus areas of endemism beginning, with 0.1˚ grid size cells, through a strict consensus or tight consensus rule [52, 53] using 5% similarity in species distribution between different sets of cells. Afterwards, we added the results obtained from analysis run with progressively larger grid cells (0.25˚, 0.3˚, 0.5˚, and 1˚)–sets resulting from larger grid cells are either the same as the smaller ones (but with coarser limits), the sums of two or more smaller ones or new areas [25]. Our complete dataset presents a number of gaps between sampled areas (Ex. S1–S8 Figs). Some of these gaps represent unsampled areas (empty cells in NDM/VNDM, S1–S8 Figs) and others represent real absences of certain species (brown cells in NDM/VNDM, S1–S8 Figs).

The second approach (ii), GIE, is based on a kernel interpolation of the species distribution, indicating level of endemism based on the area of influence of the species and their distribution overlap [18]. The spatial modeling was performed with the GIE toolbox provided by [18] in Esri ArcGis software 10.6 (http://www.esri.com). The interpolation is based on the definition of areas of influence around species centroid points. These centroid points are calculated as arithmetic means of each species sampling coordinates [54]. Five categories of influence were determined based on the maximum distance between the centroid and the furthest point of occurrence for each species. Each category defines the radius of the area of influence around each species' centroid. The "influence" of each area decreases outwards, away from the centroid, according to a Gaussian function. The overlap between the areas of influence of all species was estimated by the kernel algorithm and the results were expressed in consensus normalized raster grids, representing relative levels of endemism. Although the establishment of the limits of classes and weight that can be given to each class is in certain way arbitrary, the representation of areas of endemism as natural units with fuzzy boundaries is more compatible with the theoretical models of distribution of AoE (see [13], [4]). In this study, we performed a variation of said arbitrary units (categorization schemes) to analyze the influence they may have on AoE identification through GIE. The first analysis was obtained by analyzing the normalized data (i.e. each of the five categories in the analysis had the same influence). The second analysis was done using five categories, with the predefined weight given by the GIE tool set [18]. In this analysis, species with smaller ranges weighted more (single record species not included in the consensus), progressively and exponentially losing weight in the successive categories. Species with a single record, and therefore present in a single locality, were named as 'Single Record'.

Species ranges that delimit or influence AoE in the numerical analyses (i. e., EA and GIE) were used to delimit the Congruence Cores (CC, [25]) of AoE. A CC is an endemic area by definition, but does not define an AoE by itself. Apart from species exclusively present within a CC, there may also occur species that do not contribute to the identification of a CC in the analyses. These species occur either in more than one CC, thus are defined and classified as widespread (WS) (i.e. a WS species is a species found in at least two different CCs), or in one CC and outside of it, but never within the limits of another CC. The latter situation is the basis for delimiting a 'Maximum Region of Endemism' (MRE) [25]. Therefore a MRE is an expansion of the AoE using the distribution of species which are found inside its CC but never inside another CC. Finally, all species included in the study were classified within one of the four levels of endemicity (CC, MRE, WS, and Single Record species, modified from [25]). AoE are thus delimited with a CC and a MRE, when applied in addition to the delimitations directly achieved by the numerical methods (EA and GIE).

### Ethics statement

Field permit number: ICMBio 57798–1, ICMBio 11748–4 (Instituto Chico Mendes de Conservação da Biodiversidade (ICMBio)); EBBAut.020/2013 (Museu de Zoologia da Universidade de São Paulo (MZUSP)); 42.520/2007 (Instituto do Meio Ambiente de Santa Catarina (IMA) (former FATMA)).

## Results

### Endemicity analysis with NDM/VNDM—method (i)

Best adjustment of the points of grid origin were to -56.0 (X), -19.1 (Y). The analysis employing the 0.1˚ cell size grid, produced four sets of preliminary AoE. The consensus analysis of these sets generated the following three consensus-sets. The northernmost consensus set, named North Santa Catarina (Table 1 and S1 Fig) is located in the north-east region of the

state of Santa Catarina (Fig 3). It is defined by two species (to see species list see S1–S8 Figs) (latter defined using 0.25˚-cells as well). Consensus-set of South Santa Catarina (S2 Fig) is located in the south-east region of the state of Santa Catarina (Fig 3); it is defined by two species (three species define with 0.25˚-cells as well). The southernmost consensus-set identified is the São Francisco de Paula (S3 Fig), located in the Floresta Nacional de São Francisco de Paula, in the state of Rio Grande do Sul (Fig 3). It is defined by five species.

The analysis employing 0.25˚ cell size grid generated nine sets, and the consensus analysis generated six sets, namely the above-mentioned three sets using 0.1˚-cells (obviously differing in the size, but defined by the same species each, respectively) plus two new sets. One of these new sets is the consensus-set of the Serra dos Órgãos (S4 Fig), located in the Serra dos Órgãos, in the state of Rio de Janeiro (Fig 3). This set is defined by two species (the latter species define with 0.5˚-cells as well). The other consensus-set is the Serra do Mar de São Paulo (S5 Fig), located in the state of São Paulo (Fig 3). This set is defined by three species. The remaining sets produced would be a disjunct AoE, which overlaps with other sets previously generated (South Santa Catarina and São Francisco de Paula) (S6 Fig) and therefore presents WS species; thus, it cannot be considered an AoE.

Analysis run using 0.3˚ cell size grids generated sets of preliminary AoE all overlapping with sets generated with smaller cell size grids. These results are not shown.

The analysis employing 0.5˚ cell size grid generated 19 sets, and the consensus analysis generated five sets four of which overlap sets previously generated by the other sized grid analyses plus

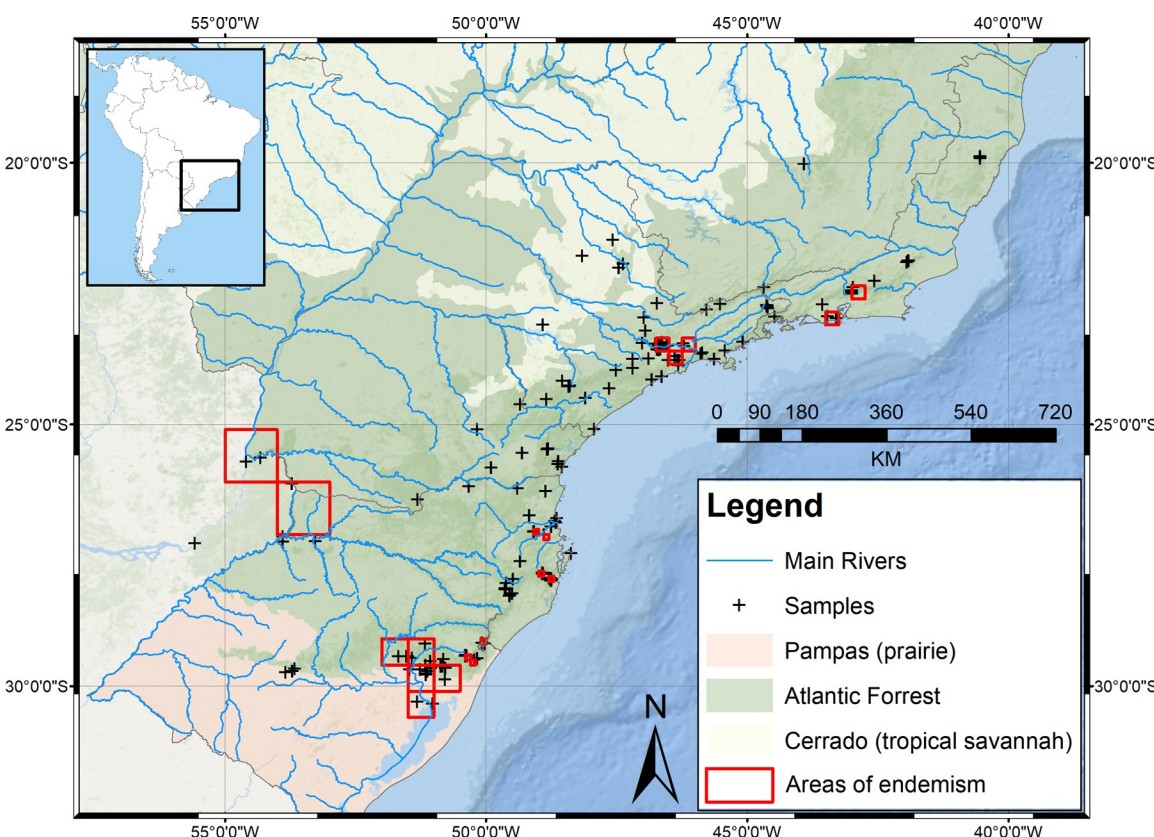

**Fig 3.** Endemicity analysis results obtained with A) 0.1˚ cell size grids. B) 0.25˚ cell size grids. C) 0.5˚ cell size grids. D) 1˚ cell size grids. Basemap source: Esri. "Topographic" [basemap]. Scale Not Given. "World Topographic Map". https://www.arcgis.com/home/item.html?id=30e5fe3149c34df1ba922e6f5bbf808f.

only one new, named Porto Alegre (S7 Fig). This consensus-set is located in the metropolitan area of Porto Alegre, in the state of Rio Grande do Sul (Fig 3), and is defined by three species

The last analysis was run with 1° cell size grid. It generated thirty three sets from which the consensus analysis resulted in five sets. Four of these sets overlap existing sets previously generated and one set is new. This new consensus-set is named Misiones (S8 Fig) and is located in the province of Misiones, Argentina (Fig 3). This set is defined by two species.

In short, the seven AoE identified through EA, named after the geographical location of their consensus-sets, are as follows: Serra dos Órgãos, Serra do Mar de São Paulo, North Santa Catarina, South Santa Catarina, São Francisco de Paula, Porto Alegre and Misiones.

## Geographical interpolation of endemism—method (ii)

Final products of this method are expressed in density surface grids. As expected, the first analysis (with normalized categories) generated an AoE as large as our sampling area housing less inclusive minor AoE (Fig 4).

The second analysis (with five categories, default weights and species with Single Record species excluded) (Fig 5) showed major congruence with the results obtained through EA. This second analysis produced seven areas. In this analysis, species with smallest ranges weighted more (single record species not included in the consensus), progressively and exponentially losing weight in the successive categories.

We therefore identify seven areas through GIE analysis. The northernmost area, named Serra dos Órgãos, is located in the Serra dos Órgãos, in the state of Rio de Janeiro. This area is defined by some undetermined level of influence from 30 species (Table 1). Moving south, the next area identified is Serra do Mar de São Paulo. It is located in the southeastern region of the state of São Paulo, and is defined by some level of influence from 28 species (Table 1). The next area identified by GIE analysis is Southern São Paulo, located further south in the state of São Paulo. It is influenced by 19 species (Table 1). The fourth area is Paraná, located around Curitiba municipality, in the state of Paraná. This area is influenced by 13 species (Table 1). Continuing south, the area of Santa Catarina is found occupying most of the eastern region of the state of Santa Catarina. This area is influenced by 77 species (Table 1). The area of São Francisco de Paula and Porto Alegre, occupies a considerable region that extends from almost the southernmost region of the state of Santa Catarina to Porto Alegre's metropolitan area. It is defined by some influence of 34 species (Table 1). The last area identified is Misiones. Located in the province of Misiones in Argentina, this area is defined by some level of influence from six species (Table 1).

GIE does not indicate objectively the endemic species supporting an AoE [59], as a high number of them influenced in each spot of the map (see above).

Summing up, the seven AoE defined through GIE analysis are Serra dos Órgãos, Serra do Mar de São Paulo, Southern São Paulo, Paraná, Santa Catarina, São Francisco de Paula and metropolitan area of Porto Alegre, and Misiones. They were named based on their geographical location (Fig 5).

## Summarized results (Fig 6)

EA identified seven AoE (Serra dos Órgãos, Serra do Mar de São Paulo, North Santa Catarina, South Santa Catarina, São Francisco de Paula, Porto Alegre and Misiones) (Fig 3). GIE identified seven AoE (Serra dos Órgãos, Serra do Mar de São Paulo, Southern São Paulo, Paraná, Santa Catarina, São Francisco de Paula and Porto Alegre and Misiones) (Fig 5). These areas can be summarized in nine AoE of terrestrial flatworms in the Southern Atlantic forest. Three of them are found by both, EA and GIE. Another four areas found by EA are also found by GIE, but in GIE they are merged into two AoE (the sums of North Santa Catarina + South

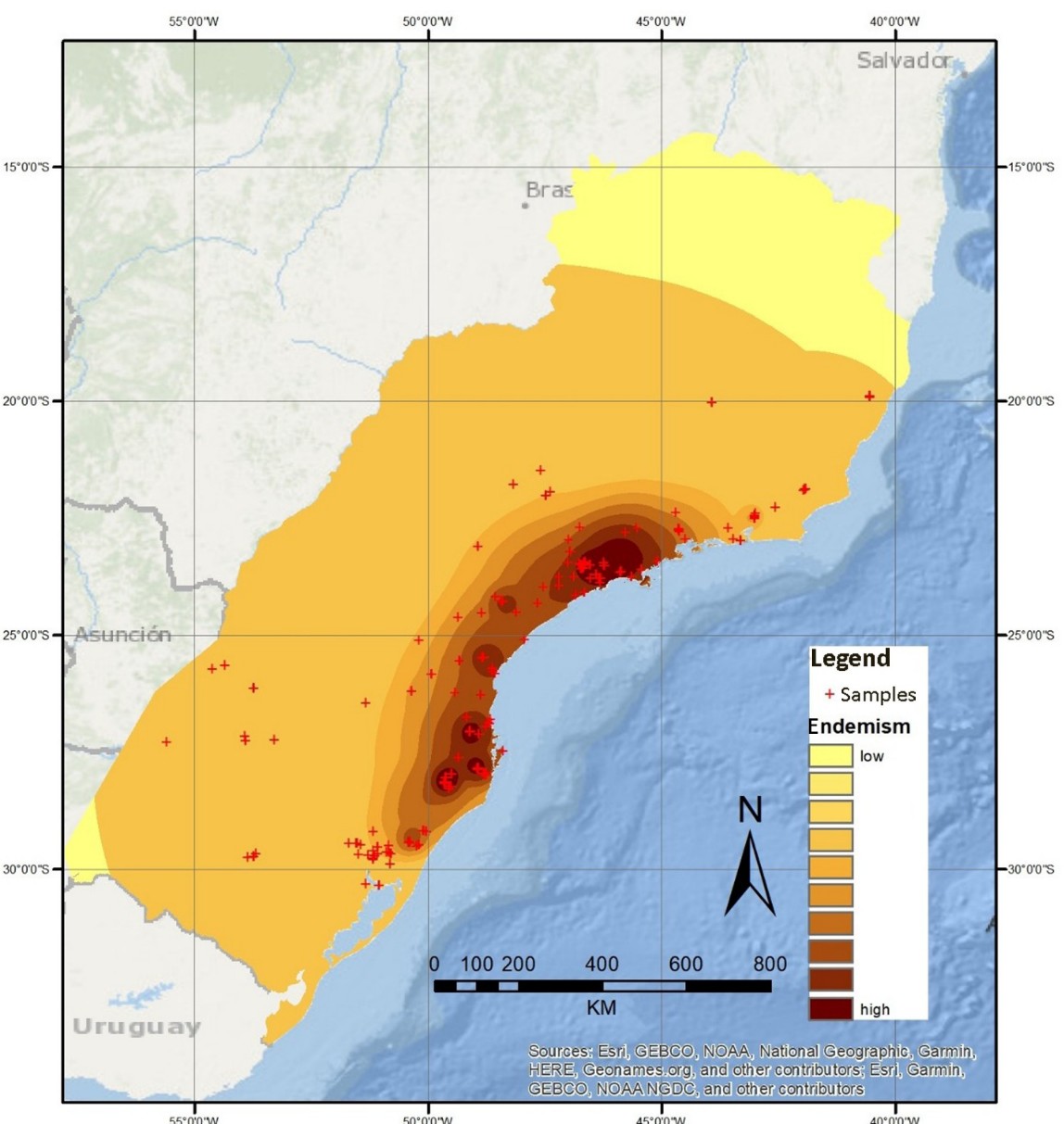

**Fig 4. Geographic interpolation of endemism results obtained with five normalized categories (each category in the analysis has the same influence) over a topographic map.** Basemap source: Esri. "Topographic" [basemap]. Scale Not Given. "World Topographic Map". https://www.arcgis.com/home/item.html?id=30e5fe3149c34df1ba922e6f5bbf808f.

Santa Catarina and Porto Alegre + Misiones). The last two AoE were found only by GIE (Paraná and Southern São Paulo).

CCs of these areas were delimited by the ranges of species that contributed to the score of EA sets, drawing a line around those species ranges (Fig 6 and S3 Table). The CCs of the two AoE found only by GIE (Paraná and Southern São Paulo) were delimited by the single localities where endemic species have congruence (in both cases, only one species have more than one record) (Fig 6).

Eleven species do not occur in any CC so they are not classified within any of the levels of endemicity (CC, MRE, WS).

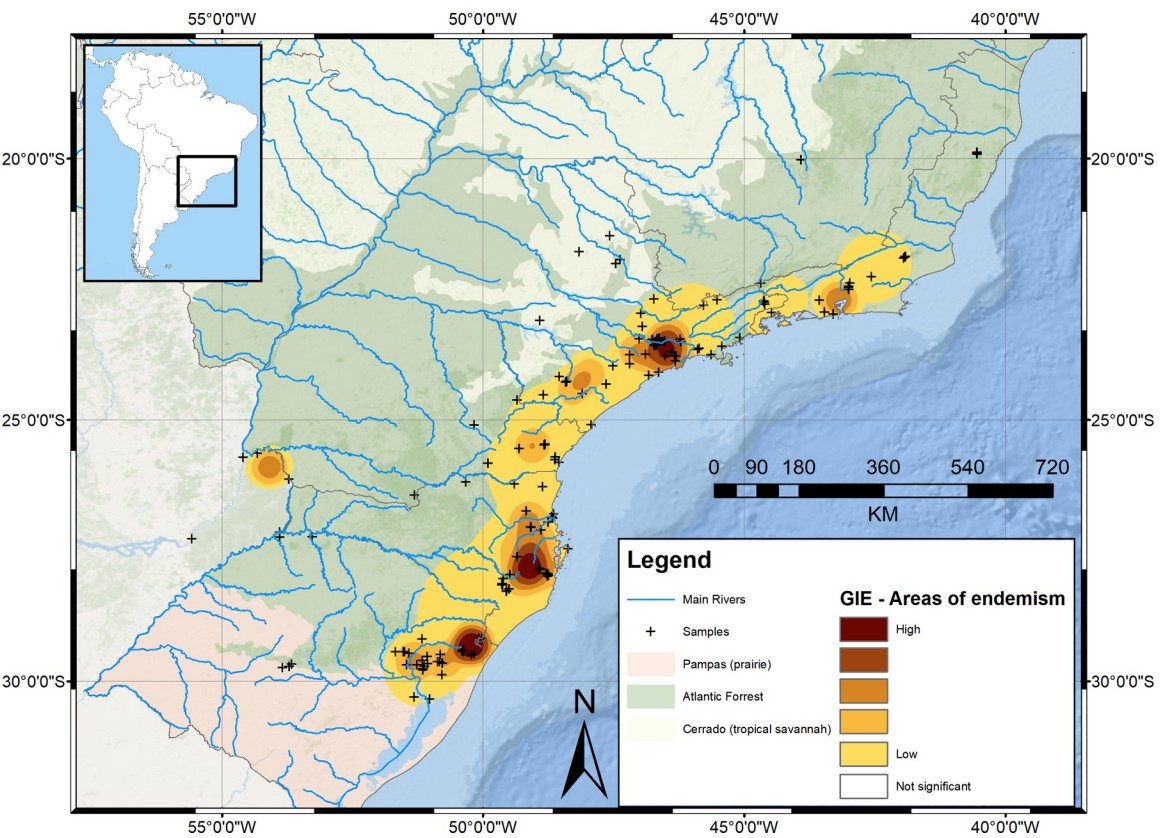

**Fig 5. Geographic interpolation of endemism results obtained with five categories, with the predefined weight given by the GIE tool set (Oliveira *et al.*, 2015) over a topographic map.** Basemap source: Esri. "Topographic" [basemap]. Scale Not Given. "World Topographic Map". https://www.arcgis.com/home/item.html?id=30e5fe3149c34df1ba922e6f5bbf808f.

The results of both numerical analyses were summarized on a map of AoE with CCs and MREs (Fig 7). ORG was found by EA and GIE and has a CC defined by two species (those that score in EA). *Geoplana notophthalma* Riester, 1938 occurs in this CC but has a record in Minas Gerais state which consequently defines a MRE of ORG. SMSP was found by EA and GIE and has a CC defined by three species (those that score in EA with 0.25˚ and 0.3˚ cells). Five species occur in the CC and widens it to a MRE (*Choeradoplana marthae*, *Choeradoplana banga*, *Obama braunsi*, *Obama evelinae*, *Obama schubarti*). Those five species delimited a set in EA with 0.5˚ cells, but actually do not have congruent ranges. This result is caused by an artifact of cell size and position for the large size of them compared to the species ranges pattern, so we decided to consider them as delimiting a MRE. PR and SSP were found only by GIE and are defined by a CC with a single locality each. SSP widens to MRE as defined by *Issoca potyra* Froehlich, 1958 and *Paraba tapira* (Froehlich, 1958) and PR widens to a MRE defined by *Notogynaphallia mourei* (Froehlich, 1956).

## Discussion

### 1. Methodological observations

As seen in previous studies, cell size and grid origin have impacts on the results obtained from an EA analysis. Large-sized cell grids generate less detailed hypotheses of species composition and area coverage of the AoE [14, 55, 56], generating a higher number of AoE, as can be seen

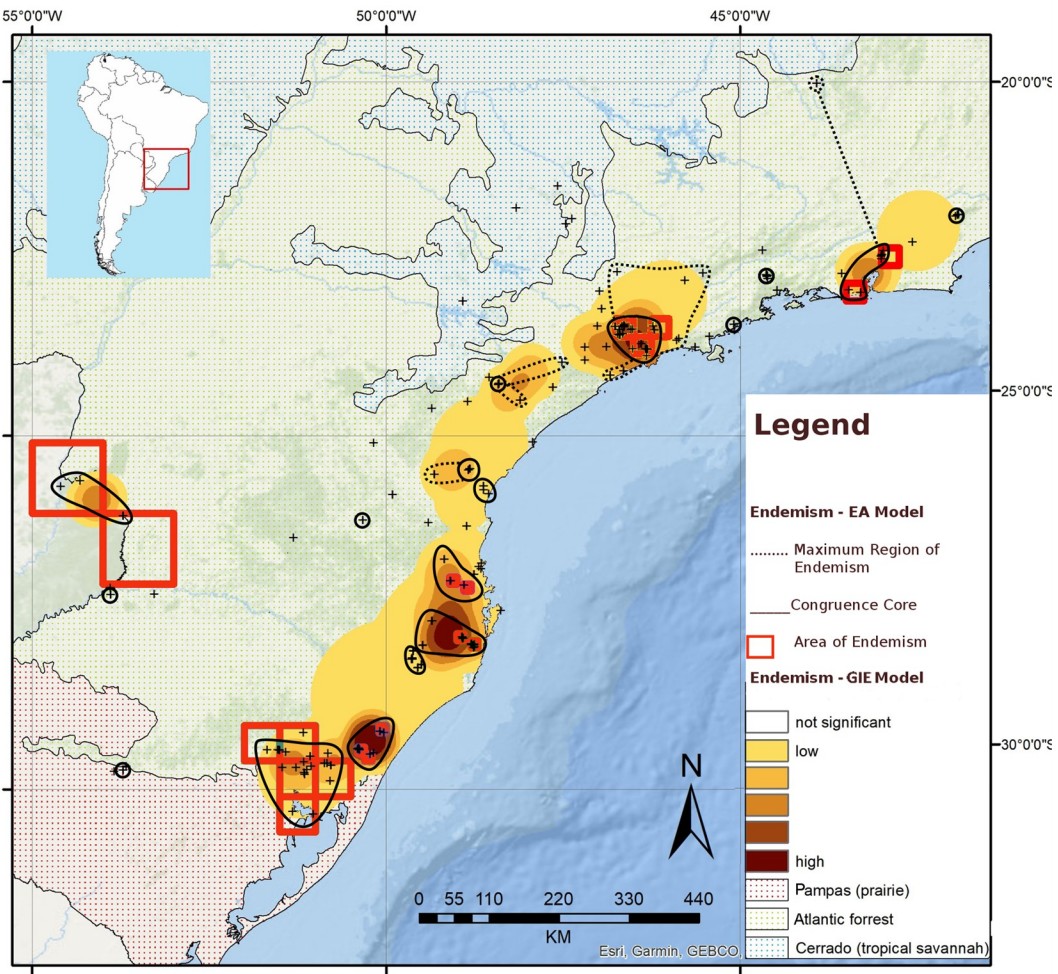

**Fig 6. Topographical map showing AoE identified in this study through the different methodologies used.** Basemap source: Esri. "Topographic" [basemap]. Scale Not Given. "World Topographic Map". https://www.arcgis.com/home/item. html?id=30e5fe3149c34df1ba922e6f5bbf808f.

in the present study. Usage of large-sized cell grids affects results, such as the collapse of two AoE into a single one [25] or the inclusion of important barriers within a single AoE [57]. On the other hand, small grids introduce false absence of species, make distributions discontinuous, and present smaller number of species per cell, making it more difficult to detect AoE and, therefore, defining fewer of them [57]. Influences derived from cell size in revealing AoE are alleviated by DaSilva's *et al.* [25] protocol. It is important to use different grid cell sizes in EA or Parsimony Analysis of Endemicity–if only one cell size grid had been used or tested with our data, some AoE would have been missed through EA.

On the other hand, GIE analysis includes another "scaling effect" similar to the cell size grid dilemma mentioned before. There are two arbitrarily chosen units in GIE analysis. One unit is the number of categories of influence to be used in the analysis. The other is the values limiting the influence of each category (the relative weight each class will have in the analysis). Tests varying the number of categories have shown that modifying this unit, will not affect significantly the results obtained via GIE analysis [18]. Although a large number of categories will definitely affect the analysis, identifying AoE with higher discontinuity and also showing

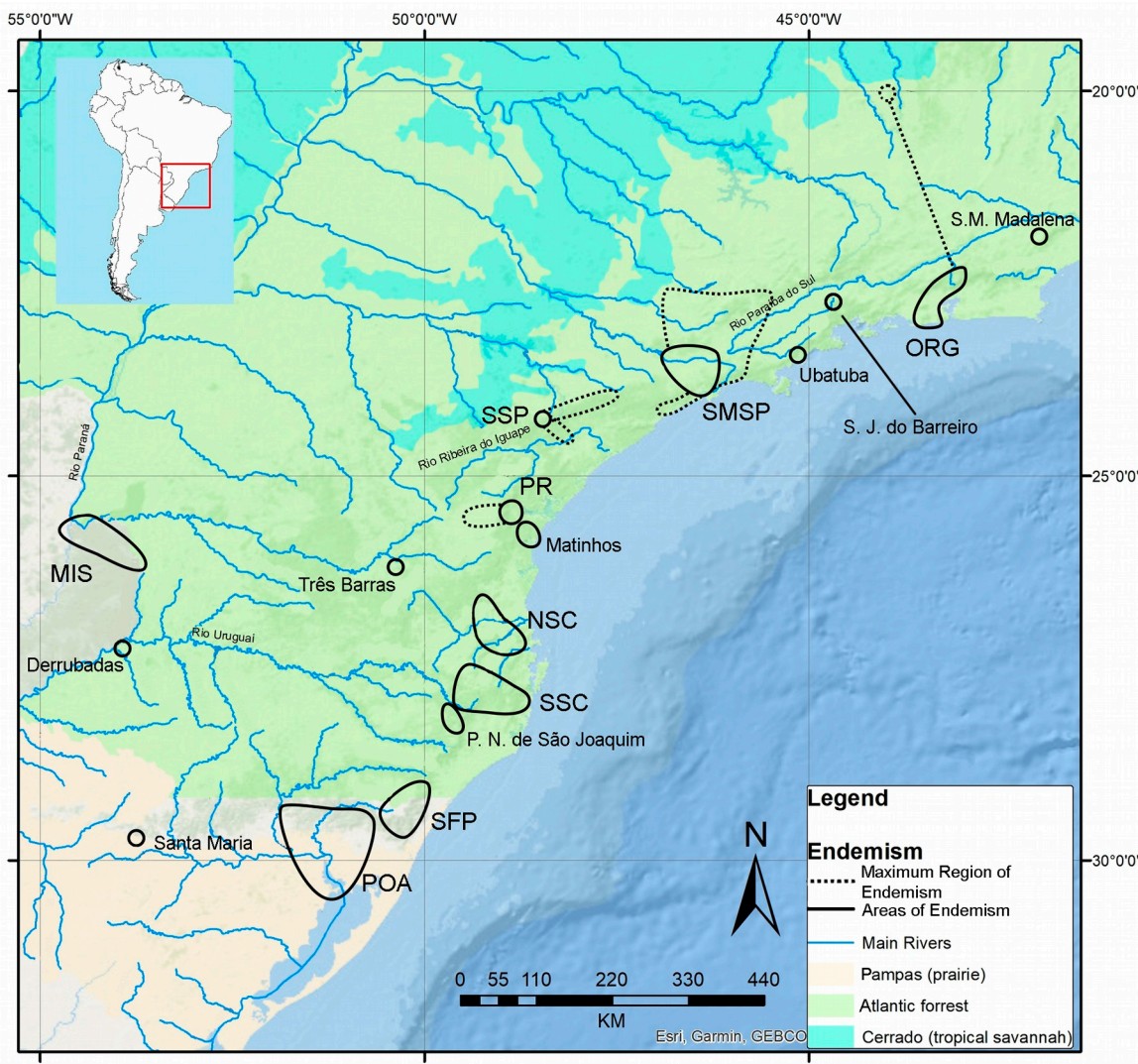

**Fig 7. Delimitation of AoE of land planarians, on a topographical map of the Atlantic forest region.** Basemap source: Esri. "Topographic" [basemap]. Scale Not Given. "World Topographic Map". https://www.arcgis.com/home/item.html?id= 30e5fe3149c34df1ba922e6f5bbf808f.

"micro areas" which sometimes are composed by a single species. This overestimates the amount of detected AoE (following Platnick's [27] definition of an AoE, a single endemic species should not be able to define AoE) (S8 Fig). Therefore the chosen categorization scheme should be tested and evaluated for every dataset analyzed in order to evade defining highly discontinuous AoE or overidentifying the number of AoE.

With the dataset analyzed with normalized weighted categories (all weights are the same for every category) (Fig 4), the obtained result is a single AoE comprising the whole study area with minor areas defined inside. This analysis is unable to identify AoE (AoE identified in this study by other GIE categorization schemes and EA) such as ORG, POA or MIS. The reason for this is the fact that the influence value of species with very restricted distributions is the same as the influence value of WS species. When trying to identify AoE, species with restricted ranges are much more indicative than WS species. Therefore, GIE analysis should be carried out with weighted categories.

The categorization scheme chosen for GIE in this study consisted of five weighted categories (default weights in the ArcGIS toolbox). Single Record species were not included in the consensus of this analysis to evade the identification of areas with one Single Record species and to try and minimize the effect of poorly sampled species on AoE identification (see [58]). As EA does not take in account single-cell sets, which would be resulted from aggregation of many Single Record species, this analysis shows high congruence with EA results.

GIE analysis overcomes several problems generated from the use of grid cells [18]. It allows the use of distributional data with gaps and offers the possibility of identifying AoE with fuzzy edges which corresponds better with the spatial and temporary reality of AoE [4, 18]. In contrast, it does not show objectively the limits of an AoE, because of this fuzzy edges.

GIE fails to provide a set of species diagnosing the identified AoE [59] apart from the "scaling effect" derived from the arbitrary units chosen for the categorization scheme. The identification of an AoE should not be possible if the list of species diagnosing that area cannot be obtained. An approximate list of species influencing a specific AoE can be obtained with GIE, but those species cannot be classified as diagnostic species because numerically it is not possible to know the influence of each species in each AoE identified.

No method, neither EA or GIE, is able to perform a complete analysis to identify AoE, due to the different flaws each methodology presents [25, 57, 60]. The analysis perform differently, offering the possibility to detect AoE based on different methodological approaches. Therefore the most efficient way to work when trying to detect AoE is through the integration of different methodologies and parameters as seen in previous studies [18, 20, 25, 42]. By working with different numerical algorithms, functions and formulas and different theoretical and methodological approaches, we are able to analyze our databases with a broader spectrum and correlate our practical results to the reality of endemic species distribution and congregation, with a higher confidence.

DaSilva's protocol facilitates the integration of different methodologies when identifying AoE. The use of a Congruence Core (CC) to delimit the main area of endemic species congruence permits the inclusion of results obtained through other methodologies efficiently and avoids the artifacts caused by grid cells, fuzzy edges or others. Defining Maximum Region of Endemism (MRE) also sets the concept that an AoE does not occupy, only, the congruent space between endemic species, but the outside areas which might represent the dynamism of said endemic species as well [25, 61]. DaSilva's protocol also maximizes EA's performance by including several steps that reduce the grid-size effect and by discarding a high number of resulted sets caused by several combinations of overlapping ranges and consequent redundancy between them. DaSilva's protocol is focused on grid-based methods, but can be an effective tool to integrate results obtained from other methodologies that obviate grids as well.

Species with a single record were included in this study, even though they are not considered in endemism analyses. In fact, EA only considers sets with two or more cells, thus a cell with many endemic species is not detected by the method as an AoE. GIE overcomes this problem, but is also unable to differentiate localities with just one species record from localities with two or more species records, as mentioned above. These limitations become evident in our study as we found eight additional localities to bear two or more endemic species recorded only once in the entire study area. From North to South, these localities are Santa Maria Madalena-RJ (two Single Record species), São José do Barreiro-SP (three Single Record species), Ubatuba-SP (four Single Record species), Matinhos-PR (four Single Record species), Três Barras-SC (two Single Record species), PN São Joaquim-SC (**31** Single Record species), Derrubadas-SC (two Single Record species) and Santa Maria-RS (two Single Record species). We believe that these species represent important biological information for planarian distribution

and diversity, and the methods should be reconsidered so the Single Record species compute for the discovery of AoE.

## 2. Inferring AoE based on land planarian distributions

Our dataset revealed nine AoE in total, plus eight localities with two or more Single Record species. Two AoE (SSP and PR) could not be detected by our EA. These AoE (SSP and PR) present an extremely high number of endemic species (19 and 18 species, respectively). These two AoE were not detected due to the fact that all except one species endemic to them are known from one locality and thus are not considered by EA, because the method only considers sets with two cells or more [25, 57].

As mentioned before, some species distributions do not completely overlap the CCs. Those species present in only one CC and outside of it will define a MRE. They can be interpreted as an extension from a CC due to ontological causes, or, as the source of an error such as undersampled regions. Most of the identified AoE do not have a MRE because planarian records are very concentrated in specific localities and regions, and those localities and regions were each delimited as an AoE (see discussion below). Only four AoE delimited MREs: ORG, SMSP, SSP and PR. Some of the species defining MREs create doubtful AoE. Such is the case of *Geoplana notophthalma* which defines a distant portion of ORG's MRE northwards, in the state of Minas Gerais. This MRE is a peculiar configuration for an AoE as AoE are defined by the congruence of only species with small distributions [62]. WS species cannot define an AoE, but *G. notophthalma* cannot be considered a WS species as it is not found in any other AoE, apart from ORG. Thus, *G. notophthalma* is found in the geographic region of Serra dos Órgãos and in Minas Gerais, and must expand ORG's AoE's borders through a MRE, thus giving rise to a disjunct AoE. Froehlich [63] suggested this species to be synonym of *O. applanata* (Graff, 1899), but the situation would not change since their distribution is the same.

Each of the six intensively sampled areas is at a distance of 120 km from the nearest intensively sampled area and, interestingly, each one was revealed as an AoE. This situation rises a question: if we had sampled between these intensively sampled areas, would we have discovered additional AoE? In other words, geographic distribution of most land planarians might occur at a scale smaller than areas with 120 km in length. General assumptions of endemism have been proposed and explained as a consequence of the ecological and physiological limitations of these organisms [43, 46, 64]. If this is true, further sampling of these organisms is promising in revealing new AoE within the study area even so because most land planarians have revealed to present a very small distributional range and each of the six areas intensely sampled turned out to be an AoE.

## 3. Causal factors of areas of endemism

In general, Serra do Mar, the mountains spreading almost along the whole geographic range herein analyzed, is considered a single area of endemism for forest-dependent vertebrate and plant taxa [22, 24, 65, 66]. However, some studies have shown a more endemic pattern of distribution for species or populations in the southeastern-south coastal mountains and adjacent lowlands of Atlantic Forest, as seen with the area of endemism of harvestmen [25] and spiders [18], specific distributional congruence of vertebrates [67], ecological regionalization of frogs [68], and phylogeographic structure of species of frogs [69–72], bees [73], vipers [74], birds [75, 76], mammals [77, 78] and harvestmen [79, 80]. The main divergence is found to the south of the state of São Paulo or close to Ribeira do Iguape river valley [6, 67, 71, 73, 74, 76], but with very different times of divergence between those taxa (~ 0.39–4.9 m.y.a.) and weak inference of causal processes.

The planarian species show very restricted ranges only compared to harvestmen species [25], or maybe to some frog or cricket taxa [81, 82]. They seem to be restricted to more humid portions of the forest on mountain slopes or adjacent regions, where the core of their areas of endemism is located, with few exceptions. Main valleys or sedimentary basins seem to be the barriers for those taxa, as described for Paraíba do Sul river and Ribeira do Iguape river [64, 67, 71, 73, 74, 76, 79, 80].

The Atlantic Forest is not physiognomically homogeneous. It presents humid forests concentrating on slopes or medium altitudes of mountains. The valleys and coastal plains are composed by a marine influenced vegetation, and interior lowlands of the south-southeastern region are composed by seasonal semideciduous forests. The endemic patterns found herein and for harvestmen show a restriction in more humid forests. This could be directly influenced by past events of forest reduction and posterior concentration in those regions, or refuges. A great body of literature from different sources and proxies continues to furnish evidence for reduction of forests in cycles during the Neogene (e.g. [70, 75, 83–85]). The high endemicity found for planarians seems to be influenced by those reductions, since PR x SC x SFP x POA and ORG x SMSP x SSP do not have clear geographic barriers. In this view, in the regions of those AoE cores, the forest should have been more humid than in lowland areas and consequently should have been maintained with few alterations during climatic fluctuations [25, 83, 86, 87].

The same should have happened to sedimentary basins wherein the great rivers flow. Those basins were originated by tectonism related with uprising of mountain ranges of Serra do Mar and Serra da Mantiqueira and subsidence of valleys since at least the Miocene [88–90]. In fact, the southeastern region of the Atlantic Forest is considered as the main rift system of eastern South America [89]. Some of those valleys started to be rain shadows as they are hidden from orographic rains and coastal humidity behind the Serra do Mar mountains (e.g. Paraíba do Sul river, [91]). Rivers originated as they flowed to the subsidence regions and most of their heads were captured from a western direction, eastward to the coast [92]. Sedimentation of those regions was affected by neotectonic reactivation [90] and rising of the amount of flowing water in more humid periods, even forming paleolakes [89] or by marine transgressions into coastal valleys [93–95]. Then, many associated events feedbacked sedimentary basins as barriers for forest-dependent taxa, avoiding the development of humid forests for long periods. They resulted in many taxa splitting in these lowlands and valley regions but in different times between each other. DaSilva *et al.* [6] proposed a model of reiterative barriers with spatial congruence in multiple times to explain those patterns.

Álvarez-Presas *et al.* [64] have shown a phylogeographic structure for *Cephaloflexa bergi* planarian populations, congruent to those main barriers, Paraíba do Sul river and Ribeira do Iguape river valleys, and a split in northern São Paulo/south Rio de Janeiro states. There is not a clear geographic barrier associated to the latter, but is congruent to ORG X SMSP AoE, cited above, the same for harvestmen AoE [25], and phylogeographic structure for *Promitobates* harvestmen [79]. The dating estimates for those main splits of *Cephaloflexa* planarian is about 8 m.y.a., in the Miocene period, near to the *Promitobates* harvestmen splits estimated in about 11 m.y.a. It is important to note that possible reductions of forest caused by climatic desiccation probably occurred previously to the renowned and more recorded Pliocene/Pleistocene fluctuations [96–98]. In spite of lack of phylogeographic structure of some populations of *Cephaloflexa*, Álvarez-Presas et al. [64] assumed that populations remained separated by those main barriers and there were few more recent dispersions.

POA and MIS areas have a different character related to their physical conditions. It is assumed that those forests have seasonal deciduous physiognomies [99] in contrast to the humid evergreen forests of the other areas of endemism. These endemisms could be associated

to those species ecological requirements or isolation from the surrounding open vegetations. However, MIS seems to be related to a separation at Paraná river or its tributary valleys, since they flow on an important sedimentary basin, as discussed above.

Based on this discussion, the hypotheses for the causes of the presented endemic patterns, inferred as speciation processes between endemic species living in those areas of endemism, should be tested or deepened by a phylogenetic framework with temporal estimation of each cladogenesis. The corroboration of the phylogeographic structure and the main populational divergence shown by Álvarez-Presas *et al.* [64] and the areas of endemism of hundreds of planarians species contribute to the understanding of the biogeographical evolution of those animals and consequently, of the humid Atlantic Forest.

## Conclusions

In this paper, we corroborate DaSilva's *et al.*, (2015) protocol as an effective tool to aid in the discovery of AoE. This protocol was used with two numerical methodologies, namely an EA (grid-based) and a GIE (centroid-based). The protocol exploits the advantages of each numerical methodology (EA allows the identification of species defining an AoE and permits a more accurate delimitation of an AoE; GIE even detects AoE consisting of a single record), whereas a number of potential AoE failed to be noticed by the numerical methods.

The AoE of land planarians discovered here are congruent with fine-scale approaches such as that addressed by DaSilva *et al.* (2015) for harvestmen. Land planarians and harvestmen are hygrophilous and forest-dependent organisms and present sensitivity to environmental changes. Presumably, due to these common physiological and ecological constraints, harvestmen and land planarians may predict, more robustly, a common history of other restricted taxa of the Atlantic Forest. Furthermore, new samplings in unexplored localities in the study area might indeed uncover new endemisms and additional AoE. Hence, these organisms might be a good model for fine-scale studies of AoE.

## Supporting information

**S1 Fig. Consensus-set North Santa Catarina obtained with NDM/VNDM.** Brown cells represent areas with an Endemicity Score of 0–1.9999 (one or no endemic species found in the cell), pink cells (pointed with arrowheads) represent areas with an Endemicity Score of 2 or higher (two or more endemic species found in the cell), empty cells represent areas where no sampling was carried out. Species giving score in pink cells: *Pasipha velina* (Froehlich, 1959) (Endemicity Index of 1.000), *Choeradoplana langi* (Graff, 1894) (Endemicity Index of 1.000).
(PDF)

**S2 Fig. Consensus-set South Santa Catarina obtained with NDM/VNDM.** Brown cells represent areas with an Endemicity Score of 0–1.9999 (one or no endemic species found in the cell), pink cells (pointed with arrowheads) represent areas with an Endemicity Score of 2 or higher (two or more endemic species found in the cell), empty cells represent areas where no sampling was carried out. Species giving score in pink cells: *Paraba tingauna* (Kishimoto & Carbayo, 2012) (Endemicity Index of 1.000), *Choeradoplana abaiba* Carbayo et al., 2017 (Endemicity Index of 1.000).
(PDF)

**S3 Fig. Consensus-set São Francisco de Paula obtained with NDM/VNDM.** Brown cells represent areas with an Endemicity Score of 0–1.9999 (one or no endemic species found in the cell), pink cells (pointed with arrowheads) represent areas with an Endemicity Score of 2 or higher (two or more endemic species found in the cell), empty cells represent areas where no

sampling was carried out. Species giving score in pink cells: *Luteostriata ceciliae* (Froehlich & Leal-Zanchet, 2003) (Endemicity Index of 0.833–1.000), *Obama josefi* (Carbayo & Leal-Zanchet, 2001) (Endemicity Index of 0.000–0.500), *Paraba franciscana* (Leal-Zanchet & Carbayo, 2001) (Endemicity Index of 0.000–0.500), *Luteostriata arturi* (Lemos & Leal-Zanchet, 2008) (Endemicity Index of 0.833–1.000), *Obama maculipunctata* Rossi *et al.*, 2016 (Endemicity Index of 0.833–1.000).
(PDF)

**S4 Fig. Consensus-set Serra dos Órgãos obtained with NDM/VNDM.** Brown cells represent areas with an Endemicity Score of 0–1.9999 (one or no endemic species found in the cell), pink cells (pointed with arrowheads) represent areas with an Endemicity Score of 2 or higher (two or more endemic species found in the cell), empty cells represent areas where no sampling was carried out. Species giving score in pink cells: *Obama fryi* (*Graff, 1899)* (Endemicity Index of 1.000), *Geoplana vaginuloides* (Darwin, 1844) (Endemicity Index of 1.000).
(PDF)

**S5 Fig. Consensus-set Serra do Mar de São Paulo obtained with NDM/VNDM.** Brown cells represent areas with an Endemicity Score of 0–1.9999 (one or no endemic species found in the cell), pink cells (pointed with arrowheads) represent areas with an Endemicity Score of 2 or higher (two or more endemic species found in the cell), empty cells represent areas where no sampling was carried out. Species giving score in pink cells: *Obama metzi* (Graff, 1899) (Endemicity Index of 0.000–0.833), *Geoplana duca* Marcus, 1951 (Endemicity Index of 0.833–1.000), *Geoplana mogi* Almeida & Carbayo, 2019 (Endemicity Index of 0.833–1.000).
(PDF)

**S6 Fig. Disjunct consensus-set which overlaps with other sets previously generated (South Santa Catarina and São Francisco de Paula) obtained with NDM/VNDM.** Brown cells represent areas with an Endemicity Score of 0–1.9999 (one or no endemic species found in the cell), pink cells (pointed with arrowheads) represent areas with an Endemicity Score of 2 or higher (two or more endemic species found in the cell), empty cells represent areas where no sampling was carried out. Species giving score in pink cells: *Choeradoplana benyai* Lemos & Leal-Zanchet, 2014 (Endemicity Index of 1.000), *Imbira sp. 1* (Endemicity Index of 1.000).
(PDF)

**S7 Fig. Consensus-set Porto Alegre obtained with NDM/VNDM.** Brown cells represent areas with an Endemicity Score of 0–1.9999 (one or no endemic species found in the cell), pink cells (pointed with arrowheads) represent areas with an Endemicity Score of 2 or higher (two or more endemic species found in the cell), empty cells represent areas where no sampling was carried out. Species giving score in pink cells: *Luteostriata abundans* (Graff, 1899) (Endemicity Index of 0.429–1.000), *Paraba gaucha* (Froehlich, 1959) (Endemicity Index of 0.800–1.000), *Pasipha hauseri* (Froehlich, 1959) (Endemicity Index of 0.700–0.833).
(PDF)

**S8 Fig. Consensus-set Misiones obtained with NDM/VNDM.** Brown cells represent areas with an Endemicity Score of 0–1.9999 (one or no endemic species found in the cell), pink cells (pointed with arrowheads) represent areas with an Endemicity Score of 2 or higher (two or more endemic species found in the cell), empty cells represent areas where no sampling was carried out. Species giving score in pink cells: *Supramontana argentina* Negrete et al., 2012 (Endemicity Index of 1.000), *Pasipha mbya* Negrete & Brusa, 2016 (Endemicity Index of 1.000).
(PDF)

**S1 Table. Geographic coordinates of the species used in the study.** Data comes from literature (reference given) and our own unpublished records (a voucher specimen is mentioned as well as the way the specimen was identified). 'Name of sp. in EA' reffers to the name of the species in the file S2 Material. Abbreviations: A1, anterior-most extremity of body; AC, copulatory apparatus; F, pharynx; Morph., morphology.
(DOCX)

**S2 Table. Records of land planarian species from literature excluded from this study and justification for such decision.**
(DOCX)

**S3 Table. Geographic distribution of the land planarian species and their classification regarding their endemicity level.** Abbreviations: CC, Congruence Core; WS, Widespread; MRE, Maximum Region of Endemism; SSC, South Santa Catarina; NSC, North Santa Catarina; Org: Serra dos Órgãos; PNSJ, Parque Nacional de São Joaquim; SFP, São Francisco de Paula; SMSP, Serra do Mar de São Paulo; SSP, Southern São Paulo; PR, Paraná; MIS, Misiones; POA, Porto Alegre; x, other cells.
(DOCX)

**S1 Material. List of references used in Fig 1.**
(DOCX)

**S2 Material. Input file of NDM/VNDM.**
(XYD)

## Acknowledgments

We thank Instituto Chico Mendes de Conservação da Biodiversidade (ICMBio), Instituto Florestal do Governo do Estado de São Paulo (IF), Museu de Zoologia da Universidade de São Paulo (MZUSP), Fundação do Meio Ambiente do Governo Estado de Santa Catarina (FATMA) for sampling licence in Parque Nacional da Serra dos Órgãos, Parque Nacional Saint Hilaire/Lange, Parque Nacional da Serra de Itajaí, Floresta Nacional de São Francisco de Paula (ICMBio); Parque Estadual Intervales (IF), Parque Estadual da Serra do Tabuleiro (FATMA). We are indebted to Ana Laura Almeida, Claudia Tangerino Olivares, Júlio Pedroni, Débora Redivo, Karine Gobetti, Leonardo Zerbone, Marília Jucá, Marta Álvarez Presas, and Welton Araujo for sampling help; Ítalo Silva de Oliveira Sousa, Geison Castro da Silveira, and Lucas Beltrami for the histological work; and K. Gobetti for the help with histological slides of E. M. Froehlich collection.

## Author Contributions

**Conceptualization:** Domingo Lago-Barcia, Marcio Bernardino DaSilva, Fernando Carbayo.

**Data curation:** Domingo Lago-Barcia, Fernando Carbayo.

**Formal analysis:** Domingo Lago-Barcia, Marcio Bernardino DaSilva, Luis Americo Conti, Fernando Carbayo.

**Funding acquisition:** Domingo Lago-Barcia, Fernando Carbayo.

**Investigation:** Domingo Lago-Barcia, Marcio Bernardino DaSilva, Luis Americo Conti, Fernando Carbayo.

**Methodology:** Domingo Lago-Barcia, Fernando Carbayo.

**Project administration:** Domingo Lago-Barcia, Fernando Carbayo.

**Resources:** Fernando Carbayo.

**Software:** Domingo Lago-Barcia, Luis Americo Conti.

**Supervision:** Fernando Carbayo.

**Validation:** Domingo Lago-Barcia, Luis Americo Conti.

**Visualization:** Domingo Lago-Barcia.

**Writing – original draft:** Domingo Lago-Barcia, Marcio Bernardino DaSilva, Fernando Carbayo.

**Writing – review & editing:** Domingo Lago-Barcia, Marcio Bernardino DaSilva, Luis Americo Conti, Fernando Carbayo.

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
