## [Decision Letter · Decision Letter 0]

6 Feb 2020

PONE-D-19-32242

Numerous areas of endemism of land planarians (Platyhelminthes: Tricladida) in the Southern Atlantic Forest as inferred from combined methods

PLOS ONE

Dear Mr. Lago-Barcia,

Thank you for submitting your manuscript to PLOS ONE. After careful consideration, we have decided that your manuscript does not meet our criteria for publication and must therefore be rejected.

Specifically:

Dear authors,

I would like to apologize for the delay. The manuscript came into my hand in December. I immediately solicited 8 reviewers and had each time an answer about Christmas holidays coming and the impossibility to review a paper in 10 days. In January, I made a second pass, allowed longer delays, and eventually obtained 3 reviews. That took time.

This is obviously an important manuscript with abundant data, and a lot of work by the authors.

Two of the reviewers are specialists of land planarians. They mentioned a number of minor or no-so-minor problems, which by themselves would justify major revision.

The third reviewer is a specialist of ecology and analysis of biodiversity and endemicity. This reviewer mentioned in the comments to the Editor "The main problem of this manuscript is that the approach, methods and question are really outdated. There are so many performing methods that could be applied, but the authors insist on this approach".

Therefore, I fear that the manuscript, even after revision, would not convince the scientific community. It is thus better to reject it and give you an opportunity to rethink the methods.

I am sorry that we cannot be more positive on this occasion, but hope that you appreciate the reasons for this decision.

Yours sincerely,

Jean-lou Justine, DrSc

Academic Editor

PLOS ONE

Reviewers' comments:

Reviewer's Responses to Questions

**Comments to the Author**

1. Is the manuscript technically sound, and do the data support the conclusions?

Reviewer #1: Yes

Reviewer #2: Yes

Reviewer #3: No

2. Has the statistical analysis been performed appropriately and rigorously? 

Reviewer #1: N/A

Reviewer #2: Yes

Reviewer #3: I Don't Know

3. Have the authors made all data underlying the findings in their manuscript fully available?

Reviewer #1: Yes

Reviewer #2: Yes

Reviewer #3: Yes

4. Is the manuscript presented in an intelligible fashion and written in standard English?

Reviewer #1: Yes

Reviewer #2: Yes

Reviewer #3: No

5. Review Comments to the Author

Reviewer #1: Dear authors,

This article is a contribution of new AoE in the Brazilian Atlantic Forest found through the use of a little-known group, the land planarians. The manuscript is perfectly understandable, written in good English and the work has been carried out with great rigor. Although there are no major errors in this paper, some changes are proposed that could help improve the manuscript. All my comments are below:

Title

In my opinion the title could be much more creative and flashier to make the article catchier for readers. Considering that PloS ONE publishes many articles, it is interesting that the title attracts the attention of the reader.

Abstract

Since abbreviations are already used in the abstract, the Areas of Endemism (AoEs) should be added at the beginning of the abstract, since the abbreviation is then used. “Areas of Endemism (AoEs) are the main study units…”

Introduction

-In the introduction an exhaustive analysis is made of all the published results on AoEs in the region, which is a little hard to follow without graphic support. It would be good to present a summary figure indicating the different areas of endemism described above on a map.

- In the same introduction, a variable number of groups of endemism are commented on according to the study analysed. It is not commented on why these numbers are so divergent, nor is it done in the discussion, which would be appreciable to comment.

- Line 91 of the introduction could be written on line 46, where the AoEs are defined, since this comment where it is now is a bit disconnected from the rest of the text.

- At the end of the introduction a short summary of the contributions of other authors so far would be positive. Why can terrestrial planarians be decisive for the analysis of AoEs in the Atlantic Forest? What is the novelty that land planarians can contribute, and other taxa already analysed, such as harvestmen, haven’t shown yet?

Material and methods

- Abbreviations should be placed before this section, otherwise, the entire text is read without knowing what the abbreviations correspond to.

-Line 114- Replace “place” by “locality”.

-Line 124- What does "sporadically in other localities between 2009 and 2018" mean? All samplings were performed on these dates? This sentence is not clear, please clarify.

- In my opinion the NDM method should be a little better explained. PloS ONE readers do not have to be specialists in the subject, and it is quite complex to understand what it is and how it works.

- Line 182- When authors refer to the manual search method, perhaps they could also name it with an abbreviation (MSM) so that it is easier to recognize in the text and in the figures.

-Line 183- A justified explanation of why this third method is used and why the two numerical methods are not enough would be lacking.

-Line 185- Have Congruence Cores (CC) been calculated with a consensus of the 3 methods? Please clarify.

-Line 189- The way in which the Maximum Region of Endemism is defined is not clear, please rewrite.

Results

-Line 230- Indicate that the results of 0.3 cell size grids are not shown.

-Line 231- Why is there so much difference in the number of initial endemic areas defined and the consensus? It goes from 19 to 5!

-Line 256- I would add the sentence "In this analysis, species with smallest ranges weighted more ..." in the text of the results, and not in the figure legend.

-Line 300- The sentence "GIE does not indicate objectively ..." should be written before, when the results of GIE are explained.

Discussion

- Given the problems that arise throughout the manuscript, closely related to the methods used to infer areas of endemism. Why have the authors not calculated the phylogenetic endemism too? Then they could have done a much more complete study. What is the goal of the study? If the authors want to define the areas of endemism for conservation purposes, more evidence is needed. Approaches that do not take phylogenetic data into account are not effective in protecting evolutionary lineages.

-Line 335- The sentence starting in this line is extremely long and is confusing, please rewrite.

-Line 341- Figure S1 is not cited in the text so far, when figures S2 to S9 have already been cited, therefore, this cannot be Figure S1, but should be S9 in the logical order of the figures.

-Line 342- Given the constraints presented by the GIE method, why have the authors only done two tests? Have not tested more parameters? Is there any statistical method to confirm the reliability of the results? Could it be applied in this case?

-Line 345- The figure quoted is wrong, I think they refer to figure 3.

-Line 370- Is that possible that the integration of the results of different methods is overestimating the number of areas of endemism?

-Line 395- The paragraph is not well understood. Delete or rewrite.

-Line 412- Have the authors analysed the copulatory apparatus of these species? Could it be true what Froehlich proposes? Maybe a phylogenetic analysis would solve this problem.

-Line 415- This whole section is a bit wordy and does not contribute anything new to the manuscript. In my opinion it can be removed up to line 429.

-Line 454- Have the authors confirmed whether the 9 areas of endemism defined coincide with any type of geographical accident? It would be interesting to overlap the layers of rainfall with the areas, to see if they coincide with really humid areas, as the authors hypothesize.

-Line 482- Replace Alvares-Prezas by Alvarez-Presas.

-Line 495- An “l” is missing in Alvarez.

Figures and Tables

-Table 1:

What is the difference between Endemic and Congruence Core (CC)? It is not indicated anywhere.

Widespread should be abbreviated (WS), like the other parameters and in the text.

- I suppose inserting the figure legends in the middle of the text is a requirement of the journal, but it is a bit weird ...

-Figure 1:

Replace Samples by Sampling sites in the figure legend.

Replace forrest by forest in the figure legend

-Figure 2:

Names in the map are not clear.

-Figure 3:

This figure is missing the Brazilian map.

This can be a supplementary figure, not necessary to be included in the main text.

Replace Legenda by Legend and amostras by sampling sites

-Figure 5:

The areas should be more highlighted, perhaps in another colour? And the names are also not very prominent. Authors could also add the number of species that define each area in brackets.

-Figure 6:

The congruence core (CC) and the areas found by MSM cannot be indicated in the same way because they are confused. Use a different colour for one of them, for example. The names of the areas found by MSM could also be added.

Replace samples by sampling sites in the legend and replace also forrest by forest.

-Figure 7:

Replace forrest by forest in the legend.

AoEs could be highlighted in a more visible way. Maybe it would be informative adding the number of species that is defining each EA.

-Figures S2 to S9:

In the figure legends authors need to explain better what is shown. Not only write the abbreviations of the names of the areas of endemism, but all the full name and the abbreviation.

These figures are uninformative. The legend needs to be better explained, what means “just ground”, for example?. Does it correspond to the "brown cells" cited in the text on line 158? Do all blank cells correspond to absences? A map of the area indicated would also be explanatory in the figures.

Why do not all the species that define the areas of endemism appear in these figures? For example, Obama assu does not appear in figure S2, and Obama nungara, Obama sp. 6 and Pasipha sp. are missing in Figure S3. Is there a justification?

-Table S1:

Delete “s” in comes. Delete “f” in refers. There are some typos in the table, such as PResas, or PNItajai (missing space).

-Table S2:

In the table legend an “a” is missing in justification. Species names and genera should be in italics.

Reviewer #2: This is an interesting study on areas of endemism using land planarians as a model. Due to land planarians’ characteristics, they are indeed an excellent group for such studies. I am not familiar with the methods used but, according to how they were explained an based on the provided references, everything seems to have been conducted correctly. I have only a few comments and suggestions that might improve the manuscript.

I think the first part of the introduction, which makes a historical analysis of Areas of Endemism proposed for the Atlantic Forest, could be more concise. Although it is an interesting presentation of previous works on the subject and in the same biome, I think it could be presented with fewer details about, for example, which Areas of Endemism each study defined.

Regarding the number and distribution of species considered in this study, I think it would be good to explain that some of the species may actually be species complexes, which is the case for species such as Luteostriata ernesti, Paraba multicolor and Paraba rubidolineata. As currently defined and based on the data shown in Supplementary Table 1, they did not help define any AoE, but once split up into separate species, they could reveal to be endemic species as well and help future studies.

I found Table 1 to be confusing in the way it is presented and, especially, the point in the text in which it appears. It would be better to refer to it at the end of the “materials and methods” section, after all its components have been explained, or even in the results section. For example, the meaning of “merged with…” found in the last column only became clear after reading the results.

I also made several suggestions to improve the language in the attached version of the manuscript.

Overall, the study is an important contribution to the knowledge of endemicity in an endangered biome, as well as an important contribution to the study of land planarians and, therefore, should be certainly published.

Reviewer #3: Dear authors,

Please find here my comments on the manuscript PONE-D-19-32242 “Numerous areas of endemism of land planarians (Platyhelminthes: Tricladida) in the Southern Atlantic Forest as inferred from combined methods”.

I had quite a hard time reading this manuscript. From a structural perspective the paper is hard to follow due to the high number of abbreviations, lack of consistent scale and delimitation of the areas on the maps, and lack of direct reading what is in the figures within the text, and many information scattered in several different supplementary figures. Language quality is very variable along the text, and requires attention. The repeated use of “said” to refer to “these” or to “already mentioned”attracted my attention, because it is not (or not commonly) employed this way in English.

After struggling to decipher some abbreviations, I found a list in page 23. More surprised I was when I found that EA and NDM (and NDM/VNDM somehow) refer to the same thing. Please reduce abbreviations as much as possible.

A specific case is Table 1). What do the abbreviations of the lines mean? Sites? Why don’t use the same abbreviations in the text?

The problem of lack of consistency in scales and the clipping in different zones makes impossible to understand figure 2. I would need to see the same background in all figures, with the presentation of different analysis as the only thing varying from one to another. The indication of the part of Brazil where they come from does not really solve the problem.

The same needs to be solved with figs 3 and 4. Why do they comprise different zones, if one specifies what is globally shown in the other?

A legend should make the figure to be read as a stand-alone information. So, it would be necessary to add the meaning of abbreviations.

A minor point concerning this is to avoid using reds and greens to show your results – please search for colorblind safe outputs so anyone can read it.

Some text reamined in Portuguese within the figures. Please check.

The huge number of supporting information in different files is also a matter of confusion and difficulties to read the manuscript as a reviewer, particularly because they are separated from legends. I guess it is a journal’s requirement (that figures are submitted without subtitles) but it would be very helpful if they were together.

General presentation apart, I have many concerns about the analysis performed here and at understanding the meaning of these areas classified.

The approach is quite old-fashioned, the literature is cited in the introduction is completely outdated – most of them date from the 80th, or 90th – even if the problem continued to be studied in many other ways. Some of these references (like reference 8) do not deal with the problem indicated.

The main problem of this approach is that areas are classified by the points of occurrence of a small sample in a huge universe of sampling possibilities. Here the authors indicate that they made a strong sampling effort at every 120km, and some gaps between these distances were covered with data from the literature. But what is in between these 120km is unknown, or poorly covered. But all biogeographer and macroecologist knows that absences might reflect three main things: lack of sample, previous occurrence followed by local extinction due to recent environmental changes, and real absences. This picture makes that classifying areas based on samples along such scattered points can lead to huge errors, and could not be useful to support any conservation strategy.

Modern biogeography relies on niche models (MaxEnt or recent Bayesian methods) to infer the distribution ranges of species. Areas of endemism are then those intersecting the highest number of the inferred range of the different species (or by the number of species desired to be defined, the information is visible and the limits can be settled by the reader of the output figure). These methods solve the problems of rough information about distribution range by coupling these rough data with fine information about the environment where the species were captured. Based on it, they identify specie’s environmental preferenda, so inferring their distribution ranges. In addition, recent studies show that they niche models can be applied to datasets with very low number of points of occurrences (even with 3, a signal can be detected), so being useful to many of the species used here. I really don’t understand why did you choose to make grids from points, or to use the Kernel interpolation when these niche models methods are so widespread, and well accepted by the scientific community?

This said, the main problem of the present study is the use of many different methods to assess these areas, the problems of changing scales and the meaning of the consensus used to select areas. As shown here it leads to such a multitude of possible outputs, and the extent they are repeatable and testable can be strongly argued. Even a manual search was necessary, because these methods are not capable to handle single localities. But what if all these localities came out of a single repeatable method in which each calibration was clearly pré-defined?

For example, one major problem evoked in the text is that areas of endemism are very dependent on the grid cell size. In a first step it results from the use of coarse grain sample to estimate (samples were mostly taken from areas about 120km apart).

Rescaling is often a good strategy to check how the patterns observed can be nested. But the choice of scales must be adequate to the distribution of the samples. You describe the way you searched for areas of endemism indicating that you searched for different grid size cells (of 0.1 º, 0.25º, 0.3º, 0.5º, and 1º) (Lines 151- 154). Considering that 1° corresponds to about 111 km near the equator, and that your grain comprised about 120km between sites, and that sites were very often much smaller than that, I wonder if refining under 1° makes sense. I guess it doesn’t, because you hardly retrieve nested areas, which is quite surprising.

Another part not explored here is that the way grid cells are placed also play an important role on the identification of areas shared by organisms or not. This could be done by shuffling the placement of the grids and trying to confirm the signal. But it was not done here.

Besides that, you are not clear when dealing with widespread species – what does widespread mean? You also about areas with disjunct distribution – not accepted in your evaluation because they were defined by widespread species.

Lines 157-159 - Some of these gaps represent unsampled areas (empty cells in NDM/VNDM, S2-S9 Figure) and others represent real absences of certain species (brown cells in NDM/VNDM, S2-S9 Figure). As indicate above absences can be due to at least three things. What do you mean by “real absences”?

6. PLOS authors have the option to publish the peer review history of their article (what does this mean?). If published, this will include your full peer review and any attached files.

Reviewer #1: Yes: Marta Álvarez-Presas

Reviewer #2: No

Reviewer #3: No

- - - - -

---

## [Author Response · Author response to Decision Letter 0]

2 Apr 2020

Dear Dr Justine and Dr. Vall-llosera Camps,

We appreciate the acceptance of our appeal request for the submission. We also thank you for the nice comments on our contribution with this manuscript, and all three referees for the careful review they made, although some of the views are not fully shared by us. 

We have revised the manuscript following most of the reviewers' suggestions and corrections and have addressed all their queries. The manuscript has been made simpler by reducing the number of abbreviations, adding some explanations to points raised by the reviewers, editing figure maps and reducing figures to the minimum needed. We believe that the manuscript has improved considerably and is now fully understandable. Please see also below our answers to the specific comments by the referees. 

We hope to have been fully complied with the editor and referee requests.

Sincerely,

The authors,

Domingo Lago-Barcia

Márcio B. DaSilva

Luis A. Conti

Fernando Carbayo

RESPONSE TO REVIEWERS (IN BLUE)

PONE-D-19-32242

Numerous areas of endemism of land planarians (Platyhelminthes: Tricladida) in the Southern Atlantic Forest as inferred from combined methods

PLOS ONE

Dear Mr. Lago-Barcia,

Thank you for submitting your manuscript to PLOS ONE. After careful consideration, we have decided that your manuscript does not meet our criteria for publication and must therefore be rejected.

Specifically:

Dear authors,

I would like to apologize for the delay. The manuscript came into my hand in December. I immediately solicited 8 reviewers and had each time an answer about Christmas holidays coming and the impossibility to review a paper in 10 days. In January, I made a second pass, allowed longer delays, and eventually obtained 3 reviews. That took time.

This is obviously an important manuscript with abundant data, and a lot of work by the authors.

Two of the reviewers are specialists of land planarians. They mentioned a number of minor or no-so-minor problems, which by themselves would justify major revision.

The third reviewer is a specialist of ecology and analysis of biodiversity and endemicity. This reviewer mentioned in the comments to the Editor "The main problem of this manuscript is that the approach, methods and question are really outdated. There are so many performing methods that could be applied, but the authors insist on this approach".

Therefore, I fear that the manuscript, even after revision, would not convince the scientific community. It is thus better to reject it and give you an opportunity to rethink the methods.

I am sorry that we cannot be more positive on this occasion, but hope that you appreciate the reasons for this decision.

Yours sincerely,

Jean-lou Justine, DrSc

Academic Editor

PLOS ONE

Reviewers' comments:

Reviewer's Responses to Questions

Comments to the Author

1. Is the manuscript technically sound, and do the data support the conclusions?

Reviewer #1: Yes

Reviewer #2: Yes

Reviewer #3: No

2. Has the statistical analysis been performed appropriately and rigorously? 

Reviewer #1: N/A

Reviewer #2: Yes

Reviewer #3: I Don't Know

3. Have the authors made all data underlying the findings in their manuscript fully available?

Reviewer #1: Yes

Reviewer #2: Yes

Reviewer #3: Yes

4. Is the manuscript presented in an intelligible fashion and written in standard English?

Reviewer #1: Yes

Reviewer #2: Yes

Reviewer #3: No

5. Review Comments to the Author

Reviewer #1. Marta Álvarez-Presas: 

Dear authors,

This article is a contribution of new AoE in the Brazilian Atlantic Forest found through the use of a little-known group, the land planarians. The manuscript is perfectly understandable, written in good English and the work has been carried out with great rigor. Although there are no major errors in this paper, some changes are proposed that could help improve the manuscript. All my comments are below:

Thanks for the nice comments. 

Title 

In my opinion the title could be much more creative and flashier to make the article catchier for readers. Considering that PloS ONE publishes many articles, it is interesting that the title attracts the attention of the reader.

We changed it to "Areas of endemism of land planarians (Platyhelminthes: Tricladida) in the Southern Atlantic Forest as inferred from combined methods"

Abstract

Since abbreviations are already used in the abstract, the Areas of Endemism (AoEs) should be added at the beginning of the abstract, since the abbreviation is then used. “Areas of Endemism (AoEs) are the main study units…”

Done.

Introduction

-In the introduction an exhaustive analysis is made of all the published results on AoEs in the region, which is a little hard to follow without graphic support. It would be good to present a summary figure indicating the different areas of endemism described above on a map.

Agree; we added the new figure 1.

- In the same introduction, a variable number of groups of endemism are commented on according to the study analysed. It is not commented on why these numbers are so divergent, nor is it done in the discussion, which would be appreciable to comment.

Numbers of areas of endemism have varied according to progressive refinement of the methods and the taxonomic group used as model, either animals or plants. It is expected that lineages coexisting in a geographical space will also share overlapped biogeographic history. Deviations from this situation may be assigned to contingencies and to the unique ecological and physiological constraints undergone by each lineage. We put this information more explicitly in the lines 97-101.

- Line 91 of the introduction could be written on line 46, where the AoEs are defined, since this comment where it is now is a bit disconnected from the rest of the text.

Done.

- At the end of the introduction a short summary of the contributions of other authors so far would be positive. Why can terrestrial planarians be decisive for the analysis of AoEs in the Atlantic Forest? What is the novelty that land planarians can contribute, and other taxa already analysed, such as harvestmen, haven’t shown yet?

As a summary, we added the following: "Refinement of the methods in the discovery of areas of endemism has accordingly produced more refined, congruent areas inferred from different taxonomic groups. General differences are related to size and shape of these areas." This sentence is placed in the lines 94-96.

Land planarians can be decisive on account of their ecological and physiological limitations as explained in the Abstract. We believe this is stated in the lines 103-109.

The novelty posed by the land planarians is the result of our study, summarized in the current title of the manuscript and discussed along the text. Studies with Opiliones have resulted in partial overlapping areas of endemism with those we discovered with planarians. This overlap should be celebrated as it provides support to the hypothesis that lineages coexisting in a geographical space also share similar biogeographic history. 

Material and methods

- Abbreviations should be placed before this section, otherwise, the entire text is read without knowing what the abbreviations correspond to.

Done. We also checked the manuscript to reduce to a minimum the number of abbreviations.

-Line 114- Replace “place” by “locality”.

Done.

-Line 124- What does "sporadically in other localities between 2009 and 2018" mean? All samplings were performed on these dates? This sentence is not clear, please clarify.

"Sporadicly" means short-termed, non previously planned samplings. These samplings are typically done during a holiday trip. We divided this sentence into two to make it clear. 

- In my opinion the NDM method should be a little better explained. PloS ONE readers do not have to be specialists in the subject, and it is quite complex to understand what it is and how it works.

The first approach (i), NDM, is based on an optimality criterion which uses species distributions to identify AoEs. It uses an heuristic algorithm to calculate an endemicity score for each cell containing endemic species. We included these sentences in lines 167-169.

- Line 182- When authors refer to the manual search method, perhaps they could also name it with an abbreviation (MSM) so that it is easier to recognize in the text and in the figures.

Done.

-Line 183- A justified explanation of why this third method is used and why the two numerical methods are not enough would be lacking.

In the revised version of the MS, we give a different treatment to the manual search and make a reflection on why the numerical methods did not retrieved these areas as AoE. 

-Line 185- Have Congruence Cores (CC) been calculated with a consensus of the 3 methods? Please clarify.

“Species ranges that delimit or influence AoEs in the numerical analyses (i. e., NDM and GIE) were used to delimit the Congruence Cores (CC, [25]) of AoEs.” Lines 208-209.

-Line 189- The way in which the Maximum Region of Endemism is defined is not clear, please rewrite.

Therefore a MRE is an expansion of the AoE using the distribution of species which are found inside its CC but never inside another CC. This sentence has been added to facilitate the understanding of MRE in lines 215-216.

Results

-Line 230- Indicate that the results of 0.3 cell size grids are not shown.

Done.

-Line 231- Why is there so much difference in the number of initial endemic areas defined and the consensus? It goes from 19 to 5!

This is because of the way NDM/VNDM works: firstly, it produces 19 preliminary sets of areas of endemism formed by different combination of endemic species. In a next step, they are transformed into a broader, consensus area were all endemic species are included.

-Line 256- I would add the sentence "In this analysis, species with smallest ranges weighted more ..." in the text of the results, and not in the figure legend.

Done.

-Line 300- The sentence "GIE does not indicate objectively ..." should be written before, when the results of GIE are explained.

Done.

Discussion

- Given the problems that arise throughout the manuscript, closely related to the methods used to infer areas of endemism. Why have the authors not calculated the phylogenetic endemism too? Then they could have done a much more complete study. What is the goal of the study? If the authors want to define the areas of endemism for conservation purposes, more evidence is needed. Approaches that do not take phylogenetic data into account are not effective in protecting evolutionary lineages.

Respectfully, we do not grasp which are the problems the reviewer refers to. The objective of the study is to contribute to the discovery of areas of endemism as inferred from the distributional data of land planarians. We believe this objective is fully achieved. Moreover, we suggested historical explanations (Discussion, lines 440-508) for the existence of these areas of endemism that may receive attention with conservation purposes because they house rare species. 

As referred in the literature cited in the Introduction [references 8, 9, 10, 11] areas of endemism themselves are a good approach to develop strategies in conservation studies. Anyway, she makes a good point suggesting an alternative approach which is out of the scope of this study since we are not focused on the discovery of evolutionary lineages. Furthermore, the systematic position of many species is uncertain (incertae sedis) and several genera are non-monophyletic. 

-Line 335- The sentence starting in this line is extremely long and is confusing, please rewrite.

We edited the sentence.

-Line 341- Figure S1 is not cited in the text so far, when figures S2 to S9 have already been cited, therefore, this cannot be Figure S1, but should be S9 in the logical order of the figures.

It is our fault. We corrected it.

-Line 342- Given the constraints presented by the GIE method, why have the authors only done two tests? Have not tested more parameters? Is there any statistical method to confirm the reliability of the results? Could it be applied in this case?

These two parameters are available in the Geographic Interpolation of Endemism toolkit, which handles only spacial parameters: the size (or radius) of each class, and the weight each of those classes will have in the Kernel interpolation. To the best of our knowledge, there are no additional parameters, nor statistical methods. 

-Line 345- The figure quoted is wrong, I think they refer to figure 3.

It is our fault. We corrected it. 

-Line 370- Is that possible that the integration of the results of different methods is overestimating the number of areas of endemism?

We do not think so. Our starting point is the definition of an area of endemism, followed by the method(s) adopted to discover them. In our study, we managed to overcome the limitations posed by each of the methods by complementing to each other. For example, EA fails to detect AoE when different endemic species have been collected in a single locality, which can be solved by integrating the results obtained from GIE. In a strict definition of an area of endemism, all methods overcome areas with unique species with a substantial congruence among their ranges. 

-Line 395- The paragraph is not well understood. Delete or rewrite.

It is our fault. We edited it.

-Line 412- Have the authors analysed the copulatory apparatus of these species? Could it be true what Froehlich proposes? Maybe a phylogenetic analysis would solve this problem.

Yes. We analysed the copulatory apparatus of these species, but have not found morphological differences that would cast doubt on Froehlich's taxonomic work. We do not have representatives of these nominal species from the type localities but certainly a phylogenetic analysis would be helpful.

-Line 415- This whole section is a bit wordy and does not contribute anything new to the manuscript. In my opinion it can be removed up to line 429.

Agree; part of the sentences were deleted..

-Line 454- Have the authors confirmed whether the 9 areas of endemism defined coincide with any type of geographical accident? It would be interesting to overlap the layers of rainfall with the areas, to see if they coincide with really humid areas, as the authors hypothesize.

We believe we have shown and discussed this coincidence in the Section 3 Causal factors of areas of endemism (lines 440-508)

-Line 482- Replace Alvares-Prezas by Alvarez-Presas.

Done.

-Line 495- An “l” is missing in Alvarez.

Done.

Figures and Tables

-Table 1:

What is the difference between Endemic and Congruence Core (CC)? It is not indicated anywhere. 

Widespread should be abbreviated (WS), like the other parameters and in the text.

We edited the text to explain what a Endemic and CC are (lines 209-210).

- I suppose inserting the figure legends in the middle of the text is a requirement of the journal, but it is a bit weird ...

Agree.

-Figure 1:

Replace Samples by Sampling sites in the figure legend.

Done.

Replace forrest by forest in the figure legend

Done.

-Figure 2:

Names in the map are not clear.

We believe this is related with the copy available for the reviewers. Original figures show readable text.

-Figure 3:

This figure is missing the Brazilian map.

This can be a supplementary figure, not necessary to be included in the main text.

Replace Legenda by Legend and amostras by sampling sites

Done. 

-Figure 5:

The areas should be more highlighted, perhaps in another colour? And the names are also not very prominent. Authors could also add the number of species that define each area in brackets.

Done.

-Figure 6:

The congruence core (CC) and the areas found by MSM cannot be indicated in the same way because they are confused. Use a different colour for one of them, for example. The names of the areas found by MSM could also be added.

Replace samples by sampling sites in the legend and replace also forrest by forest.

We edited this and the other figure maps. 

-Figure 7:

Replace forrest by forest in the legend.

Done.

AoEs could be highlighted in a more visible way. Maybe it would be informative adding the number of species that is defining each EA.

We believe that the Table 1 achieves this purpose. 

-Figures S2 to S9:

In the figure legends authors need to explain better what is shown. Not only write the abbreviations of the names of the areas of endemism, but all the full name and the abbreviation.

These figures are uninformative. The legend needs to be better explained, what means “just ground”, for example?. Does it correspond to the "brown cells" cited in the text on line 158? Do all blank cells correspond to absences? A map of the area indicated would also be explanatory in the figures.

Why do not all the species that define the areas of endemism appear in these figures? For example, Obama assu does not appear in figure S2, and Obama nungara, Obama sp. 6 and Pasipha sp. are missing in Figure S3. Is there a justification?

Agree. "Just ground" and color of the cells are default outputs of the software NDM. We edited the text legends. 

-Table S1:

Delete “s” in comes. Delete “f” in refers. There are some typos in the table, such as PResas, or PNItajai (missing space).

Done.

-Table S2:

In the table legend an “a” is missing in justification. Species names and genera should be in italics.

Done.

Reviewer #2: 

This is an interesting study on areas of endemism using land planarians as a model. Due to land planarians’ characteristics, they are indeed an excellent group for such studies. I am not familiar with the methods used but, according to how they were explained an based on the provided references, everything seems to have been conducted correctly. I have only a few comments and suggestions that might improve the manuscript.

I think the first part of the introduction, which makes a historical analysis of Areas of Endemism proposed for the Atlantic Forest, could be more concise. Although it is an interesting presentation of previous works on the subject and in the same biome, I think it could be presented with fewer details about, for example, which Areas of Endemism each study defined.

We edited the text to try and reduce its length and added a map (Figure 1) summarizing the text.

Regarding the number and distribution of species considered in this study, I think it would be good to explain that some of the species may actually be species complexes, which is the case for species such as Luteostriata ernesti, Paraba multicolor and Paraba rubidolineata. As currently defined and based on the data shown in Supplementary Table 1, they did not help define any AoE, but once split up into separate species, they could reveal to be endemic species as well and help future studies.

Good point. Correct species identification is fundamental for inferring areas of endemism and we took this into account when compiling distributional data of the species from literature. The species mentioned can represent unsolved species complexes. Future taxonomic work could disentangle this situation. We believe that the areas of endemism discovered in this study would not be affected by eventual taxonomic changes of those species because each area of endemism is grounded on a number of species. 

I found Table 1 to be confusing in the way it is presented and, especially, the point in the text in which it appears. It would be better to refer to it at the end of the “materials and methods” section, after all its components have been explained, or even in the results section. For example, the meaning of “merged with…” found in the last column only became clear after reading the results.

Agree. The table was moved to the end of Materials and Methods. 

I also made several suggestions to improve the language in the attached version of the manuscript.

Many thanks for your suggestions and corrections in the attached version; most of them were followed. 

Overall, the study is an important contribution to the knowledge of endemicity in an endangered biome, as well as an important contribution to the study of land planarians and, therefore, should be certainly published.

Thanks for the nice comment.

Reviewer #3: 

Dear authors,

Please find here my comments on the manuscript PONE-D-19-32242 “Numerous areas of endemism of land planarians (Platyhelminthes: Tricladida) in the Southern Atlantic Forest as inferred from combined methods”.

I had quite a hard time reading this manuscript. From a structural perspective the paper is hard to follow due to the high number of abbreviations, lack of consistent scale and delimitation of the areas on the maps, and lack of direct reading what is in the figures within the text, and many information scattered in several different supplementary figures. Language quality is very variable along the text, and requires attention. The repeated use of “said” to refer to “these” or to “already mentioned”attracted my attention, because it is not (or not commonly) employed this way in English.

After struggling to decipher some abbreviations, I found a list in page 23. More surprised I was when I found that EA and NDM (and NDM/VNDM somehow) refer to the same thing. Please reduce abbreviations as much as possible.

We apologize for the additional attention needed to follow abbreviations in the text. In the revised version, we tried to reduce them to a minimum. It was our fault to synonymize NDM with EA. The text should read EA to mean a method named Endemicity Analysis, NDM is a software performing Endemicity Analysis. VNDM is software built in NDM for visualizing maps. 

We also checked the maps to show them at the same scale and with the same ground whenever possible. We also omitted figures not needed to understand our results, so that the reader can easily focus on the main points of the study. 

A specific case is Table 1). What do the abbreviations of the lines mean? Sites? Why don’t use the same abbreviations in the text?

We edited the Table to make it clear. 

The problem of lack of consistency in scales and the clipping in different zones makes impossible to understand figure 2. I would need to see the same background in all figures, with the presentation of different analysis as the only thing varying from one to another. The indication of the part of Brazil where they come from does not really solve the problem.

The same needs to be solved with figs 3 and 4. Why do they comprise different zones, if one specifies what is globally shown in the other?

We changed the size of the figures to the same scale. 

A legend should make the figure to be read as a stand-alone information. So, it would be necessary to add the meaning of abbreviations.

Done.

A minor point concerning this is to avoid using reds and greens to show your results – please search for colorblind safe outputs so anyone can read it.

Some text reamined in Portuguese within the figures. Please check.

Done.

The huge number of supporting information in different files is also a matter of confusion and difficulties to read the manuscript as a reviewer, particularly because they are separated from legends. I guess it is a journal’s requirement (that figures are submitted without subtitles) but it would be very helpful if they were together.

Agree. Whenever possible, we avoid the use of colors and reduced the number of supporting information to a minimum needed to understand the results. It is a journal’s requirement to include the figure legend’s in the manuscript text and not in the figure file. 

General presentation apart, I have many concerns about the analysis performed here and at understanding the meaning of these areas classified.

The approach is quite old-fashioned, the literature is cited in the introduction is completely outdated – most of them date from the 80th, or 90th – even if the problem continued to be studied in many other ways. Some of these references (like reference 8) do not deal with the problem indicated. The main problem of this approach is that areas are classified by the points of occurrence of a small sample in a huge universe of sampling possibilities. Here the authors indicate that they made a strong sampling effort at every 120km, and some gaps between these distances were covered with data from the literature. But what is in between these 120km is unknown, or poorly covered. But all biogeographer and macroecologist knows that absences might reflect three main things: lack of sample, previous occurrence followed by local extinction due to recent environmental changes, and real absences. This picture makes that classifying areas based on samples along such scattered points can lead to huge errors, and could not be useful to support any conservation strategy.

Modern biogeography relies on niche models (MaxEnt or recent Bayesian methods) to infer the distribution ranges of species. Areas of endemism are then those intersecting the highest number of the inferred range of the different species (or by the number of species desired to be defined, the information is visible and the limits can be settled by the reader of the output figure). These methods solve the problems of rough information about distribution range by coupling these rough data with fine information about the environment where the species were captured. Based on it, they identify specie’s environmental preferenda, so inferring their distribution ranges. In addition, recent studies show that they niche models can be applied to datasets with very low number of points of occurrences (even with 3, a signal can be detected), so being useful to many of the species used here. I really don’t understand why did you choose to make grids from points, or to use the Geographic Interpolation of Endemism when these niche models methods are so widespread, and well accepted by the scientific community?

With all respect, we disagree with your view. Historical biogeography is a dynamic discipline with several approaches and methods, one of them is on areas of endemism. Its concepts, theoretical approaches and methodological discussions have been extensively discussed in updated high impact articles. It is related to speciation, geographic barriers, common biogeographic patterns and so on. Areas of endemism may be defined as an area with two or more endemic species living in them, presenting substantial congruence among their range limits (Platnick 1991). Specific methods explore exhaustive distributional data of species to discover areas of endemism. In agreement with this, the methods and findings of our study are sound. 

The fact that the method is old-fashioned (a viewpoint which we do not agree with; Geographic Interpolation of Endemism was firstly proposed in 2015 in a paper published in PlosOne, (Oliveira et al, 2015) does not imply that is does not produce scientifically robust results. The novelty per se posed by new methodological approaches is not a reason for neglecting current methods. A method should be abandoned if its scientific incorrectness is proved. This is not the case. To the best of our knowledge, there is no literature that invalidates our approach. This point also meets Journal's Criteria for publication; Plos One is not specifically concerned with novelty, but with robust results: "Experiments, statistics, and other analyses are performed to a high technical standard and are described in sufficient detail" (please find the remaining 6 criteria at: https://journals.plos.org/plosone/s/criteria-for-publication).

The reviewer recommends to adopt different approaches such as niche modeling. She/He makes a good point. Although it is evidently out of the scope of this study, it would be interesting to compare our results with those produced with niche modeling. Niche modeling is constrained by the abundance of distributional data of each species. We handled 570 records of 270 species. Among them, 230 species are only known from 1-2 records (85% of the total species), 17 species by 3 records (6%) and 23 species by more than three records (9%). We have no expertise in niche modeling, but from reviewer’s wording, only 15% of the species (those with 3 or more records) would be suitable for that approach. It implies a data loss of 85% of the species.

Furthermore, the climatic suitability of taxa does not necessarily represent the actual ranges of species distribution. In previous studies, physical parameters did not explain the distribution of land planarian species (Antunes et al., 2012; Álvarez-Presas et al., 2018); alternatively, historical processes have been claimed to explain current distribution of these species (Álvarez-Presas et al., 2018), probably as a consequence of the ecological and physiological limitations of these organisms. Therefore, niche modeling will seemingly explore just a fraction of our data and add noise to our distributional information, which is exclusively based on true records taken in the field and the present work explicitly aims to delimit areas of endemism. In our view, the large amount of true spatial data of so many species of land planarians constitute a great virtue in our study. 

Antunes et al., 2012. Habitat structure, soil properties, and food availability do not predict terrestrial flatworms occurrence in Araucaria Forest sites. Pedobiologia 55: 25–31, https://doi.org/10.1016/j.pedobi.2011.09.010

Álvarez-Presas et al., 2018. Hidden diversity in forest soils: Characterization and comparison of terrestrial flatworm’s communities in two national parks in Spain. Ecology and Evolution 1–15, DOI: 10.1002/ece3.4178

Oliveira U, Brescovit AD, Santos AJ. 2015. Delimiting Areas of Endemism through Kernel Interpolation. PLoS ONE. 10(1) DOI:10.1371/journal.pone.0116673

Platnick NI. On areas of endemism. Aust. Syst. Bot. 1991;4: 11-12

Finally, with respect to the reviewer’s comment above ("Some of these references (like reference 8) do not deal with the problem indicated"), we reply as follows: REF 8 (Vane-Wright et al., 1991) was mentioned here especifically because of authors’view: "McNaughton (1989) has observed that we have to 'determine what should be conserved and how it is to be conserved. A criticalplaces strategy.., could accomplish this objective'. McNaughton is an ecologist, and his 'critical places' refer to representative ecosystems. As systematists we think instead of areas of endemism, or critical faunas and floras for particular taxonomic groups." (p. 242, 2nd paragraph).

This said, the main problem of the present study is the use of many different methods to assess these areas, the problems of changing scales and the meaning of the consensus used to select areas. As shown here it leads to such a multitude of possible outputs, and the extent they are repeatable and testable can be strongly argued. Even a manual search was necessary, because these methods are not capable to handle single localities. But what if all these localities came out of a single repeatable method in which each calibration was clearly pré-defined?

For example, one major problem evoked in the text is that areas of endemism are very dependent on the grid cell size. In a first step it results from the use of coarse grain sample to estimate (samples were mostly taken from areas about 120km apart).

Rescaling is often a good strategy to check how the patterns observed can be nested. But the choice of scales must be adequate to the distribution of the samples. You describe the way you searched for areas of endemism indicating that you searched for different grid size cells (of 0.1 º, 0.25º, 0.3º, 0.5º, and 1º) (Lines 151- 154). Considering that 1° corresponds to about 111 km near the equator, and that your grain comprised about 120km between sites, and that sites were very often much smaller than that, I wonder if refining under 1° makes sense. I guess it doesn’t, because you hardly retrieve nested areas, which is quite surprising.

Another part not explored here is that the way grid cells are placed also play an important role on the identification of areas shared by organisms or not. This could be done by shuffling the placement of the grids and trying to confirm the signal. But it was not done here.

As we highlighted along the text (lines 333-408) each method we used presents limitations in finding areas of endemism. The combined methods avoid constraints derived from sampling bias, a situation that will virtually never be solved. We believe we discussed it extensively in the Discussion section and provide support to combine these methods. The multiple outputs are just rough results in the form of preliminary maps that after treatment produce consensus areas of endemism. Detailed material and methods and Supporting information are intended to allow reproducibility of our results. We also do not intend to consider the matter closed but to contribute to the debate of discovery and test of congruence of areas of endemism in the Atlantic forest. In the future, further samplings in non explored areas will help test the existence of new areas of endemism or to reshape the ones we discovered in this study. 

We rethought the “manual search method”, because of its subjectivity, and decided to remove it as a step from our methods, focusing on NDM and GIE results. 

The problem related to grid size was properly approched by Oliveira et al (2015), who developed a method that does not require grids, and DaSilva et al (2016), who developed a protocol that details the reasoning behind scale in the area of endemism searching. We believe that the use of both approaches (including NDM which is a commonly used method to delimit areas of endemism), and the integration of them is a virtue of the present work.

“We adjusted the points of grid origin to the default position provided by NDM and to 3 other origins to test for eventual hidden AoE derived from position...” We added this information in the text in lines 173-174.

Besides that, you are not clear when dealing with widespread species – what does widespread mean? You also about areas with disjunct distribution – not accepted in your evaluation because they were defined by widespread species.

Widespread species are the species found in two or more areas of endemism. Subsequently they are not endemic to any area. They are maintained in the analyses but they do not contribute to -nor weight against- the delimitation of an AoE. We edited the text (lines 212-213) to make it clear. 

Disjunct AoE can be accepted. The problem arises when the disjunct AoE is defined by widespread species, i.e., species occurring in two separated areas of endemism. An AoE cannot be identified by a widespread species due to the definition of a widespread species (mentioned above). A species cannot be found in two different AoE as it invalidates its status of endemicity. This approach is based on the work of DaSilva et al (2016) and earlier literature cited in this paper.

Lines 157-159 - Some of these gaps represent unsampled areas (empty cells in NDM/VNDM, S2-S9 Figure) and others represent real absences of certain species (brown cells in NDM/VNDM, S2-S9 Figure). As indicate above absences can be due to at least three things. What do you mean by “real absences”?

"Unsampled areas" are areas where no sampling has been conducted. "Real absences" are sampled areas where certain species have not been found. We made it clear in the new version of the manuscript. We do not know to what extent a species not observed in a sampled area is really absent. Most probably, the number of species really inhabiting an area is larger than the number of species actually found, even in the six intensely sampled areas. This bias is virtually unavoidable. A general overview of our results shows that as we make progress in the study of areas of endemism in the Atlantic forest, more refined and non-conflicting areas are being discovered. The new figure 1 included in the manuscript might help to show it.

---

## [Decision Letter · Decision Letter 1]

26 Jun 2020

Areas of endemism of land planarians (Platyhelminthes: Tricladida) in the Southern Atlantic Forest

PONE-D-19-32242R1

Dear Dr.  Lago-Barcia

Your manuscript has been judged scientifically suitable for publication and will be formally accepted for publication once it meets all outstanding technical requirements.

Kind regards,

Tunira Bhadauria, Ph.D.

Academic Editor

PLOS ONE

and

Michael A Chadwick, PhD

Academic Editor

PLOS ONE

Journal Requirements:

1.) Your ethics statement must appear in the Methods section of your manuscript. If your ethics statement is written in any section besides the Methods, please move it to the Methods section and delete it from any other section. Please also ensure that your ethics statement is included in your manuscript, as the ethics section of your online submission will not be published alongside your manuscript.

2.) We note that Figures in your submission contain [map/satellite] images which may be copyrighted. All PLOS content is published under the Creative Commons Attribution License (CC BY 4.0), which means that the manuscript, images, and Supporting Information files will be freely available online, and any third party is permitted to access, download, copy, distribute, and use these materials in any way, even commercially, with proper attribution. For these reasons, we cannot publish previously copyrighted maps or satellite images created using proprietary data, such as Google software (Google Maps, Street View, and Earth). For more information, see our copyright guidelines: http://journals.plos.org/plosone/s/licenses-and-copyright.We require you to either

(1) present written permission from the copyright holder to publish these figures specifically under the CC BY 4.0 license, or

(2) remove the figures from your submission:

1. You may seek permission from the original copyright holder of Figure(s) [#] to publish the content specifically under the CC BY 4.0 license. We recommend that you contact the original copyright holder with the Content Permission Form (http://journals.plos.org/plosone/s/file?id=7c09/content-permission-form.pdf) and the following text:

The following resources for replacing copyrighted map figures may be helpful:USGS National Map Viewer (public domain): http://viewer.nationalmap.gov/viewer/

Additional Editor Comments (optional):

I believe that all the corrections made by the authors in the first review have improved the manuscript enough to make it final and suitable for publication.

Reviewers' comments:

Reviewer's Responses to Questions

**Comments to the Author**

1. If the authors have adequately addressed your comments raised in a previous round of review and you feel that this manuscript is now acceptable for publication, you may indicate that here to bypass the “Comments to the Author” section, enter your conflict of interest statement in the “Confidential to Editor” section, and submit your "Accept" recommendation.

Reviewer #1: All comments have been addressed

Reviewer #4: All comments have been addressed

2. Is the manuscript technically sound, and do the data support the conclusions?

Reviewer #1: Yes

Reviewer #4: Yes

3. Has the statistical analysis been performed appropriately and rigorously? 

Reviewer #1: N/A

Reviewer #4: I Don't Know

4. Have the authors made all data underlying the findings in their manuscript fully available?

Reviewer #1: Yes

Reviewer #4: Yes

5. Is the manuscript presented in an intelligible fashion and written in standard English?

Reviewer #1: Yes

Reviewer #4: Yes

6. Review Comments to the Author

Reviewer #1: I believe that the authors have made a great effort to comply with and correct or answer all the proposals made by the reviewers. I have attached the pdf with some minor comments on it. The manuscript has improved a lot, in my opinion, and that's why I think it should be published. This is an important contribution to the better knowledge of AoE in the Atlantic Forest, a field that has been little explored.

Reviewer #4: The general structure of the work, as well as the starting hypotheses, the objectives set and the conclusions reached, seem to me very accurate.

I am not familiar with the methods used by the authors to identify AoE, for this reason I will not evaluate this methodology. The ms is written in an understandable way, making it possible to follow the line of reasoning even for non-specialists in these methods (like me).

In my opinion, it is very interesting to carry out this type of study with terrestrial planarians. A little-known and little-studied zoological group, but of great importance in terrestrial ecosystems, especially in the tropical and subtropical regions.

7. PLOS authors have the option to publish the peer review history of their article (what does this mean?). If published, this will include your full peer review and any attached files.

Reviewer #1: Yes: Marta Álvarez-Presas

Reviewer #4: Yes: Eduardo Mateos

---

## [Editor Report · Acceptance letter]

7 Jul 2020

PONE-D-19-32242R1 

Areas of endemism of land planarians (Platyhelminthes: Tricladida) in the Southern Atlantic Forest 

Dear Dr. Lago-Barcia:

I'm pleased to inform you that your manuscript has been deemed suitable for publication in PLOS ONE. Congratulations! Your manuscript is now with our production department. 

Kind regards, 

on behalf of

Dr. Tunira Bhadauria 

Academic Editor

PLOS ONE